# IDENTIFYING WEIGHT-VARIANT LATENT CAUSAL MODELS

## ABSTRACT

The task of causal representation learning aims to uncover latent higher-level causal representations that affect lower-level observations. Identifying true latent causal representations from observed data, while allowing instantaneous causal relations among latent variables, remains a challenge, however. To this end, we start from the analysis of three intrinsic indeterminacies in identifying latent space from observations: transitivity, permutation indeterminacy, and scaling indeterminacy. We find that transitivity acts as a key role in impeding the identifiability of latent causal representations. To address the unidentifiable issue due to transitivity, we introduce a novel identifiability condition where the underlying latent causal model satisfies a linear-Gaussian model, in which the causal coefficients and the distribution of Gaussian noise are modulated by an additional observed variable. Under some mild assumptions, we can show that the latent causal representations can be identified up to trivial permutation and scaling. Furthermore, based on this theoretical result, we propose a novel method, termed Structural caUsAl Variational autoEncoder (SuaVE), which directly learns latent causal representations and causal relationships among them, together with the mapping from the latent causal variables to the observed ones. We show that SuaVE learns the true parameters asymptotically. Experimental results on synthetic and real data demonstrate the identifiability and consistency results and the efficacy of SuaVE in learning latent causal representations.

## 1 INTRODUCTION

While there is no universal formal definition, one widely accepted feature of disentangled representations (Bengio et al., 2013) is that a change in one dimension corresponds to a change in one factor of variation in the underlying model of the data, while having little effect on others. The underlying model is rarely available for interrogation, however, which makes learning disentangled representations challenging. Several excellent works for disentangled representation learning have been proposed that focus on enforcing independence over the latent variables that control the factors of variation (Higgins et al., 2017; Chen et al., 2018; Locatello et al., 2019; Kim & Mnih, 2018; Locatello et al., 2020). In many applications, however, the latent variables are not statistically independent, which is at odds with the notion of disentanglement above, *i.e.*, foot length and body height exhibit strong positive correlation in the observed data (Träuble et al., 2021).

Causal representation learning avoids the aforementioned limitation, as it aims to learn a representation that exposes the unknown high-level causal structural variables, and the relationships between them, from a set of low-level observations (Schölkopf et al., 2021). Unlike disentangled representation learning, it identifies the possible causal relations among latent variables. In fact, disentangled representation learning can be viewed as a special case of causal representation learning where the latent variables have no causal influences (Schölkopf et al., 2021). One of the most prominent additional capabilities of causal representations is the ability to represent interventions and to make predictions regarding such interventions (Pearl, 2000), which enables the generation of new samples that do not lie within the distribution of the observed data. This can be particularly useful to improve the generalization of the resulting model. Causal representations also enable answering counterfactual questions, *e.g.*, would a given patient have suffered heart failure if they had started exercising a year earlier (Schölkopf et al., 2021)?

Despite its advantages, causal representation learning is a notoriously hard problem—without certain assumptions, identifying the true latent causal model from observed data is generally not possible. There are three primary approaches to achieve identifiability: 1) adapting (weakly) supervised methods with given latent causal graphs or/and labels (Kocaoglu et al., 2018; Yang et al., 2021; Von Kügelgen et al., 2021; Brehmer et al., 2022), 2) imposing sparse graphical conditions, *e.g.*, with bottleneck graphical conditions (Adams et al., 2021; Xie et al., 2020; Lachapelle et al., 2021), 3) using temporal information (Yao et al., 2021; Lippe et al., 2022). A brief review is provided in Section 2. For the supervised approach, when labels are known, the challenging identifiability problem in latent space has been transferred to an identifiability problem in the observed space, for which some commonly-used functional classes have been proven to be identifiable (Zhang & Hyvarinen, 2012; Peters et al., 2014). Given latent causal graphs overly depends on domain knowledge. For the second approach, many true latent causal graphs do not satisfy the assumed sparse graph structure. The temporal approach is only applicable when temporal information or temporal intervened information among latent factors is available.

In this work, we explore a new direction in the identifiability of latent causal representations, by allowing causal influences among latent causal variables to change, motivated by recent advances in nonlinear ICA (Hyvarinen et al., 2019; Khemakhem et al., 2020). (Hyvarinen et al., 2019; Khemakhem et al., 2020) have shown that with an additional observed variable $\mathbf{u}$ to modulate latent independent variables, the latent independent variables are identifiable. Then a question naturally arises: with causal relationships among latent variables, what *additional assumptions* are required for the identifiability? To answer this question, we start from the analysis of three intrinsic indeterminacies in latent space (see Section 3): transitivity, permutation indeterminacy, and scaling indeterminacy, which further give rise to the following insights. 1) Transitivity is the scourge of identifiablity of the latent causal model. 2) Permutation indeterminacy means the recovered latent variables can have arbitrary permutation of the underlying orders, due to the flexibility in latent space. This nature enables us to enforce the learned causal representations in correct causal orders regularized by a predefined a directed acyclic supergraph, avoiding troublesome directed acyclic graph (DAG) constraint. 3) Scaling and permutation indeterminacy only allow recovering latent causal variables up to permutation and scaling, not the exact values. To overcome the transitivity challenge we model the underlying causal representation with weight-variate linear Gaussian models, where both the weights (i.e., causal coefficients) and the mean and variance of the Gaussian noise are modulated by an additional observed variable $\mathbf{u}$ (see Section 4). With these assumptions, we can show that the latent causal representations can be recovered up to a trivial permutation and scaling. The key to the identifiability is that the causal influences (weights) among the latent causal variables are allowed to change. Intuitively, the changing causal influences enable us to obtain interventional observed data, which enables the identifiability of the latent causal variables. Based on this result, in Section 5, we further propose a novel method, Structural caUsAl Variational autoEncoder (SuaVE), for learning latent causal representations with consistency guarantee. Section 6 verifies the efficacy of the proposed approach on both synthetic and real fMRI data.

## 2 RELATED WORK

Due to the challenges of identifiability in causal representation learning, most existing works handle this problem by imposing assumptions. We thus give a brief review of the related work on this basis.

**(Weakly) Supervised Causal Representation Learning** Approaches falling within this class assume known latent causal graphs or labels. CausalGAN (Kocaoglu et al., 2018) requires a-priori knowledge of the structure of the causal graph of latent variables, which is a significant practical limitation. CausalVAE (Yang et al., 2021) needs additional labels to supervise the learning of latent variables. Such labels are not commonly available, however, and manual labeling can be costly and error-prone. Von Kügelgen et al. (2021) use a known but non-trivial causal graph between content and style factors to study self-supervised causal representation learning. Brehmer et al. (2022) learn causal representation in a weakly supervised setting whereby they assume access to data pairs representing the system before and after a randomly chosen unknown intervention.

**Sparse Graphical Structure** Most recent progress in identifiability focuses on sparse graphical structure constraints (Silva et al., 2006; Shimizu et al., 2009; Anandkumar et al., 2013; Frot et al., 2019; Cai et al., 2019; Xie et al., 2020; 2022). Adams et al. (2021) provided a unifying viewpoint of

this approach whereby a sparser model that fits the observation is preferred. However, they primarily consider linear relations between latent causal variables, and between latent causal variables and observed variables, which are often violated in real-world applications. Lachapelle et al. (2021) handles nonlinear causal relation between latent variables by assuming special sparse graphical structures. However, many latent causal graphs in reality may be more or less arbitrary, beyond a purely sparse graph structure. In contrast, our work assumes a function class of latent variables, and does not restrict the graph structure over them.

**Temporal Information**  The temporal constraint that the effect cannot precede the cause has been used repeatedly in latent causal representation learning (Yao et al., 2021; Lippe et al., 2022; Yao et al., 2022). For example, Yao et al. (2021) recover latent causal variables and the relations between them using Variational AutoEncoders and enforcing constraints in causal process prior. Lippe et al. (2022) learn causal representations from temporal sequences, which requires the underlying causal factors to be intervened. All of these works can be regarded as special cases of exploring the change of causal influences among latent variables in time series data. The approach proposed is more general since the observed auxiliary variable $\mathbf{u}$ could represent time indices, domain indices, or almost any additional or side information.

Besides, Kivva et al. (2021) considers a nonlinear setting for latent causal discovery. It assumes discrete latent causal variables that are rendered identifiable by a mixture oracle, while in this work we consider continues latent causal variables.

## 3  THREE INDETERMINACIES IN LATENT CAUSAL MODELS

In this section, we first build a connection between nonlinear ICA and causal representation learning, by exploiting the correspondence between the independence of latent variables in nonlinear ICA and the independence of latent noise variables in causal representations. We then consider three indeterminacies in latent space: transitivity, permutation indeterminacy, and scaling indeterminacy, and analyse their impact on identifiability.

### 3.1  RELATING CAUSAL REPRESENTATION LEARNING WITH IDENTIFIABLE NONLINEAR ICA

Causal representation learning aims to uncover latent higher-level causal representations that can explain lower-level raw observations with a nonlinear mapping. Specifically, we assume that the observed variables $\mathbf{x}$ are influenced by latent causal variables $z_i$, and the causal structure among $z_i$ can be any directed acyclic graph (which is unknown). For each latent causal variable $z_i$, there is a corresponding latent noise variable $n_i$, as shown in Figure 1, which represents some unmeasured factors that influence $z_i$. The latent noise variables $n_i$ are assumed to be independent [1] with each other, *conditional on the observed variable* $\mathbf{u}$, in a

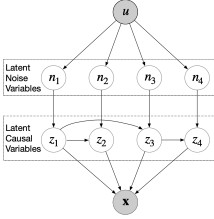

Figure 1: Causal representation with $\mathbf{u}$.

causal system (Peters et al., 2017), so it is natural to leverage recent progress in nonlinear ICA (Hyvarinen et al., 2019; Khemakhem et al., 2020), which has shown that the independent latent noise variables $n_i$ are identifiable under relatively mild assumptions, if they are modulated by an additionally observed variable $\mathbf{u}$. For example, if $\{n_i\}$ are Gaussian distributed with their mean and variance modulated by an observed variable $\mathbf{u}$, and the mapping from $\mathbf{n}$ to $\mathbf{z}$ is invertible. Taking one step further, our goal is to recover the latent causal variables $z_i$. However, with the assumptions for the identifiability of latent noise variables $n_i$ from nonlinear ICA, it is still insufficient to identify the latent causal variables $z_i$. The reason of such non-identifiability will be given in Section 3.2. Given this fact, a further question is what additional conditions are needed to recover the latent causal variables $z_i$. The corresponding identifiability conditions will be given in Section 4.

---

[1] For convenience in the later parts, with a slight abuse of definition, independent $n_i$ means that $n_i$ are mutually independent conditional on the observed variable $\mathbf{u}$.

### 3.2 TRANSITIVITY: THE CHALLENGE OF IDENTIFYING LATENT CAUSAL REPRESENTATIONS

Even with the identifiable $n_i$, it is impossible to identify the latent causal variables $z_i$ without additional assumptions. To interpret this point, for simplicity, let us only consider the influences of $z_1$ and $z_2$ on $\mathbf{x}$ in Figure 1. According to the graph structure in Figure 1, assume that $z_1 := n_1$, $z_2 := z_1 + n_2$ and $\mathbf{x} := \mathbf{f}(z_1, z_2) + \varepsilon$ (case 1). We then consider the graph structure shown in the left column of Figure 2, where the edge $z_1 \rightarrow z_2$ has been removed, and assume that $z_1 := n_1$, $z_2 := n_2$ and $\mathbf{x} := \mathbf{f} \circ \mathbf{g}(z_1, z_2) + \varepsilon$ where $\mathbf{g}(z_1, z_2) = [z_1, z_1 + z_2]$ (case 2). Interestingly, we find that the causal models in case 1 and case 2 generate the same observed data $\mathbf{x}$, which implies that there are two different causal models to interpret the same observed data. Clearly, $z_2$ in both the two equivalent causal structures are different, and thus $z_2$ is unidentifiable.

Similarly, we can also cut the edge $z_1 \rightarrow z_3$ in Figure 1, obtaining another equivalent causal graph as shown in the right of Figure 2. That is, we can have two different $z_3$ to interpret the same observed data, and thus $z_3$ is unidentifiable. Since there exists many different equivalent causal structures, the latent causal variables in Figure 1 is unidentifiable. Such result is because that the effect of $z_1$ on $z_2$ (or $z_3$) in latent space can be 'absorbed' by the nonlinear function from $\mathbf{z}$

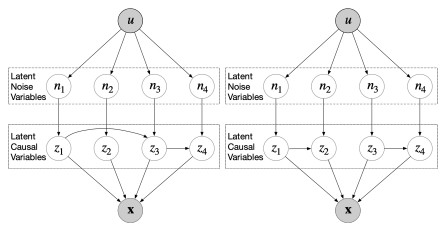

Figure 2: Two equivalent graph structures.

to $\mathbf{x}$. We term this indeterminacy transitivity in this work. We will show how to handle this challenge in Section 4. Before that, we introduce the other two indeterminacies in latent space, which assists in understanding the proposed identifiablity result.

### 3.3 SCALING INDETERMINACY IN RECOVERING LATENT CAUSAL REPRESENTATIONS

The scaling indeterminacy of latent causal variables is also an intrinsic indeterminacy in latent space. Again, for simplicity, we only consider the influence of $z_1$ and $z_2$ on $\mathbf{x}$ in Figure 1, and assume that $z_1 := n_1$, $z_2 := z_1 + n_2$ and $\mathbf{x} := \mathbf{f}(z_1, z_2) + \varepsilon$. Under this setting, if the value of $z_1$ is scaled by $s$, $e.g.$ $s \times z_1$, we can easily obtain the same observed data $\mathbf{x}$ by: 1) letting $z_2 := \frac{1}{s} \times z_1 + n_2$ and 2) $\mathbf{x} := \mathbf{f} \circ \mathbf{g}(z_1, z_2) + \varepsilon$ where $\mathbf{g}(z_1, z_2) = [s \times z_1, z_1]$. This indeterminacy is because the scaling of the latent variables $z_i$ can be 'absorbed' by the nonlinear function from $\mathbf{z}$ to $\mathbf{x}$ and the causal functions among the latent causal variables. Therefore, without additional information to determine the values of the latent causal variables $z_i$, it is only possible to identify the latent causal variable up to scaling, not exactly recovering the values. In general, this scaling does not affect identifying the causal structure among the latent causal variables. We will further discuss this point in Section 4.

### 3.4 PERMUTATION INDETERMINACY IN RECOVERING LATENT CAUSAL REPRESENTATIONS

Due to the nature of ill-posedness, latent causal representation learning suffers from permutation indeterminacy, where the recovered latent causal representations have an arbitrary permutation. For example, assume that the underlying (synthetic) latent causal representations are corresponding to the size, color, shape, and location of an object, and we obtain the recovered latent

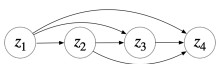

Figure 3: A Causal Supergraph.

causal representations $z_1, z_2, z_3, z_4$. Permutation indeterminacy means that we can not ensure that the recovered $z_1$ represents which specified semantic information, $e.g.$, the size or the color. Therefore, without additional information to determine the order of the recovered latent causal representations $z_i$, it is only possible to identify the latent causal representations up to permutation.

Although the flexibility and ambiguity in latent space causes permutation indeterminacy bringing some troubles, they also bring a benefit for learning latent causal representations with a straightforward way to handle the causal order on latent causal representations. That is, we can always pre-define a causal order $z_1, z_2, z_3, z_4$, without specified semantic information for the nodes. Based on this, we

can further pre-define a causal supergraph as depicted by Figure 3 [2]. As a result, the four predefined nodes $z_1, z_2, z_3, z_4$ are enforced to learn four corresponding latent variables in the correct causal order. For example, assume that a correct underlying causal order is: size $\succ$ color $\succ$ shape $\succ$ location. Since $z_1$ is the first node in the predefined causal supergraph, $z_1$ is enforced to the semantic information of the first node in the correct underlying causal order, *e.g.*, the size. Similarly, $z_2$ is enforced to the semantic information of the second node in the correct underlying causal order, *e.g.*, the color. The proposed causal supergraph can naturally ensure a directed acyclic graph estimation in learning causal representation, avoiding DAG constraints (such as the one proposed in Zheng et al. (2018)). We will further show how to implement it in the proposed method for learning latent causal representations in Section 5.

# 4 IDENTIFIABLE LATENT CAUSAL REPRESENTATIONS WITH WEIGHT-VARIANT LINEAR GAUSSIAN MODELS

As discussed in Section 3.2, the key factor that impedes identifiable causal representations is the transitivity in latent space. Note that the transitivity is because the causal influences among the latent causal variables may be 'absorbed' by the nonlinear mapping from latent variables $\mathbf{z}$ to the observed variable $\mathbf{x}$. To address this issue, we allow causal influences among latent causal variables to be modulated by an additionally observed variable $\mathbf{u}$, represented by the red edge in Figure 4. Intuitively, variant causal influences among latent causal variables cannot be 'absorbed' by an invariant nonlinear mapping from $\mathbf{z}$ to $\mathbf{x}$, resulting in identifiable causal representations. We assume $\mathbf{x}$ and $\mathbf{z}$ satisfy the following causal models:

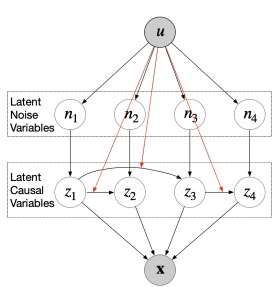

Figure 4: The proposed Latent Causal Model.

$$n_i :\sim \mathcal{N}(\beta_{i,1}(\mathbf{u}), \beta_{i,2}(\mathbf{u})), \tag{1}$$

$$z_i := \boldsymbol{\lambda}_i^T(\mathbf{u})(\mathbf{z}) + n_i, \tag{2}$$

$$\mathbf{x} := \mathbf{f}(\mathbf{z}) + \boldsymbol{\varepsilon} \tag{3}$$

where

- each noise term $n_i$ is Gaussian distributed with mean $\beta_{i,1}(\mathbf{u})$ and variance $\beta_{i,2}(\mathbf{u})$; both $\beta_{i,1}$ and $\beta_{i,2}$ can be nonlinear mappings. Moreover, the distribution of $n_i$ is modulated by the observed variable $\mathbf{u}$.

- In Eq. (2), $\boldsymbol{\lambda_i}(\mathbf{u})$ denote the vector corresponding to the causal weights from $\mathbf{z}$ to $z_i$, *e.g.* $\boldsymbol{\lambda_i}(\mathbf{u}) = [\lambda_{1,i}(\mathbf{u}), \lambda_{2,i}(\mathbf{u}), ..., \lambda_{n,i}(\mathbf{u})]$, and each $\lambda_{j,i}$ could be a nonlinear mapping.

- In Eq. (3), $\mathbf{f}$ denote a nonlinear mapping, and $\boldsymbol{\varepsilon}$ is independent noise with probability density function $p_{\boldsymbol{\varepsilon}}(\boldsymbol{\varepsilon})$.

In Eq. (2), we assume the latent causal model of each $z_i$ satisfies a linear causal model with the causal weights being modulated by $\mathbf{u}$, and we call it *weight-variant linear Gaussian model*. Therefore, $p(\mathbf{z}|\mathbf{u})$ satisfies the following multivariate Gaussian distribution:

$$p_{(\boldsymbol{\lambda}, \boldsymbol{\beta})}(\mathbf{z}|\mathbf{u}) = \mathcal{N}(\boldsymbol{\mu}, \Sigma), \tag{4}$$

---

[2]Here a causal supergraph is defined a directed acyclic graph in which there is a directed edge for any two variables.

with the mean $\boldsymbol{\mu}$ and the covariance matrix $\Sigma$ computed by the following recursion relations (Bishop & Nasrabadi, 2006; Koller & Friedman, 2009):

$$\mu_i = \sum_{j \in \mathrm{pa}_i} \lambda_{i,j}(\mathbf{u})\mu_j + \beta_{i,1}(\mathbf{u}), \quad \Sigma_{i,i} = \sum_{j \in \mathrm{pa}_i} \lambda_{i,j}^2(\mathbf{u})\Sigma_{j,j} + \beta_{i,2}(\mathbf{u}),$$
$$\Sigma_{i,j} = \sum_{k \in \mathrm{pa}_j} \lambda_{j,k}(\mathbf{u})\Sigma_{i,k}, \quad for \quad i \neq j, \tag{5}$$

where $\mathrm{pa}_i$ denotes the parent nodes of $z_i$.

Furthermore, this multi-variate Gaussian can be re-formulated with the following exponential family:

$$p_{(\mathbf{T},\boldsymbol{\lambda},\boldsymbol{\beta})}(\mathbf{z}|\mathbf{u}) = \frac{1}{Z(\boldsymbol{\lambda},\boldsymbol{\beta},\mathbf{u})} \exp\left(\mathbf{T}^T(\mathbf{z})\boldsymbol{\eta}(\mathbf{u})\right), \tag{6}$$

where the natural parameter $\boldsymbol{\eta}(\mathbf{u}) = [\Sigma^{-1}\boldsymbol{\mu}; -\frac{1}{2}\mathrm{vec}(\Sigma^{-1})]$, $Z(\boldsymbol{\lambda},\boldsymbol{\beta},\mathbf{u})$ denotes the normalizing constant, and the sufficient statistic $\mathbf{T}(\mathbf{z}) = [\mathbf{z}; \mathrm{vec}(\mathbf{z}\mathbf{z}^T)]$ ('vec' denotes the vectorization of a matrix). We further denote by $\mathbf{T}_{\min}$ the minimal sufficient statistic and by $k$ its dimension, with $2n \leq k \leq n + (n * (n+1))/2$. In particular, $k = 2n$ corresponds to the case when $\mathbf{T}_{\min}(\mathbf{z}) = [z_1, z_2, ..., z_n, z_1^2, z_2^2, ..., z_n^2]$, or in other words, there is no edges among the latent variables $\mathbf{z}$, while $k = n + (n * (n+1))/2$ corresponds to a full-connected causal graph over $\mathbf{z}$. So with different causal structures over $\mathbf{z}$, the dimension $k$ may vary. For the graph structure in Figure 1, $\mathbf{T}_{\min}(\mathbf{z}) = [z_1, z_2, z_3, z_4, z_1^2, z_2^2, z_3^2, z_4^2, z_1 z_2, z_1 z_3, z_3 z_4]$ and $k = 11$.

The following theorem shows that under certain assumptions on the nonlinear mapping $\mathbf{f}$ and the variability of $\mathbf{u}$, the latent variable $\mathbf{z}$ can be identifiability up to trivial permutation and scaling.

**Theorem 4.1** *Suppose latent causal variables $\mathbf{z}$ and the observed variable $\mathbf{x}$ follow the generative models defined in Eq. 1- Eq. 3, with parameters $(\mathbf{f}, \boldsymbol{\lambda}, \boldsymbol{\beta})$. Assume the following holds:*

*(i) The set $\{\mathbf{x} \in \mathcal{X} | \varphi_\varepsilon(\mathbf{x}) = 0\}$ has measure zero (i.e., has at most countable number of elements), where $\varphi_\varepsilon$ is the characteristic function of the density $p_\varepsilon$.*

*(ii) The function $\mathbf{f}$ in Eq. 3 is bijective.*

*(iii) There exist $k + 1$ distinct points $\mathbf{u}_0, \mathbf{u}_1, ..., \mathbf{u}_k$ such that the matrix*
$$\mathbf{L} = (\boldsymbol{\eta}(\mathbf{u}_1) - \boldsymbol{\eta}(\mathbf{u}_0), ..., \boldsymbol{\eta}(\mathbf{u}_k) - \boldsymbol{\eta}(\mathbf{u}_0)) \tag{7}$$
*of size $k \times k$ is invertible.*

*(iv) For each $i, j$, $\lambda_{i,j}(\mathbf{0}) = 0$,*

*then the true latent causal variables $\mathbf{z}$ are related to the estimated latent causal variables $\hat{\mathbf{z}}$ by the following relationship:*
$$\mathbf{z} = \mathbf{P}\hat{\mathbf{z}} + \mathbf{c},$$
*where $\mathbf{P}$ denotes the permutation matrix with scaling, $\mathbf{c}$ denotes a constant vector.*

Intuitively, the identifiability result benefits from the weight-variant linear Gaussian model in Eq. 2. The auxiliary observed variable $\mathbf{u}$ modulates the variant weights among latent causal variables, which can be regarded as an intervention and contributes to the identifiability, so a wrong causal structure over the latent variables will violate the assumed model class. As a challenging problem, causal representation learning is known to be generally impossible without further assumptions. The proposed weight-variant linear Gaussian model explores the possibility of identifiable causal representation learning, without any special causal graphical structure constraints. The proof is given in Appendix.

The above theorem has shown that the latent causal variables can be identified up to trivial permutation and linear scaling. With this result, identifiability of causal structure in *latent space* reduces to the identifiability of the causal structure in *observed space*. With the help of recent progress in causal discovery from heterogeneous data (Huang* et al., 2020), the following corollary shows that the causal structure among latent variables $\mathbf{z}$ is also identifiable up to the Markov equivalence class. Moreover, although there exists scaling indeterminacy for the recovered latent variables as stated in Theorem 4.1, it does not affect the identifiability of the causal structure.

**Corollary 4.2** *Suppose latent causal variables $\mathbf{z}$ and the observed variable $\mathbf{x}$ follow the generative models defined in Eq. 1- Eq. 3, and that the conditions in Theorem 4.1 hold. Then, under the assumption that the joint distribution over $\mathbf{z}$ and $\mathbf{u}$ is Markov and faithful to $G_{\mathbf{z} \cup \mathbf{u}}$, the acyclic causal structure among latent variables $\mathbf{z}$ can be identified up to the Markov equivalence class of $G_{\mathbf{z}}$, where $G_{\mathbf{z} \cup \mathbf{u}}$ denotes the causal graph over $\mathbf{z} \cup \mathbf{u}$ and $G_{\mathbf{z}}$ denotes the causal graph over $\mathbf{z}$.*

# 5 LEARNING CAUSAL REPRESENTATIONS WITH WEIGHT-VARIANT LINEAR GAUSSIAN MODELS

Based on the identifiable results above, in this section, we propose a structural causal variational autoencoder (SuaVE) to learn latent causal representations. We first propose a structural causal model prior relating to the proposed weight-variant linear Gaussian model. We then show how to incorporate the proposed prior into the traditional VAE framework (Kingma & Welling, 2013a). We finally give consistency result for the proposed SuaVE.

## 5.1 PRIOR ON LEARNING LATENT CAUSAL REPRESENTATIONS

Since we can always have a super causal graph for the causal structure among $\mathbf{z}$ as shown in Figure 3 as discussed in 3.4, the corresponding weight matrix $\boldsymbol{\lambda} = [\boldsymbol{\lambda}_1(\mathbf{u}), ..., \boldsymbol{\lambda}_n(\mathbf{u})]$ can be constrained as a upper triangular matrix with zero-value diagonal elements. As a result, the causal model for the latent causal variables, e.g., Eq. 1 and Eq. 2, can be reformulated with the following probabilistic model.

$$p(\mathbf{z}|\mathbf{u}) = p(z_1|\mathbf{u}) \prod_{i=2}^{n} p(z_i|\mathbf{z}_{<i}, \mathbf{u}) = \prod_{i=1}^{n} \mathcal{N}(\mu_{z_i}, \delta_{z_i}^2), \tag{8}$$

where

$$\mu_{z_i} = \sum_{i' < i} \lambda_{i,i'}(\mathbf{u})(z_{i'}) + \beta_{i,1}(\mathbf{u}), \qquad \delta_{z_i}^2 = \beta_{i,2}(\mathbf{u}). \tag{9}$$

The proposed prior naturally ensures a directed acyclic graph estimation, avoiding additional DAG constraint. In contrast, traditional methods, *i.e.* CausalVAE (Yang et al., 2021), employ a relaxed DAG constraint proposed by Zheng et al. (2018) to estimate the causal graph, which may result in a cyclic graph estimation due to the inappropriate setting of the regularization hyper parameter.

## 5.2 SUAVE FOR LEARNING LATENT CAUSAL REPRESENTATIONS

The nature of the proposed prior in Eq. 8 and Eq. 9 gives rise to the following variational posterior:

$$q(\mathbf{z}|\mathbf{x}, \mathbf{u}) = q(z_1|\mathbf{x}, \mathbf{u}) \prod_{i=2}^{n} q(z_i|\mathbf{z}_{<i}, \mathbf{x}, \mathbf{u}) = \prod_{i=1}^{n} \mathcal{N}(\mu'_{z_i}, \delta'^2_{z_i}), \tag{10}$$

where:

$$\mu'_{z_i} = \sum_{i' < i} \lambda'_{i,i'}(\mathbf{u}) z_{i'} + \beta'_{i,1}(\mathbf{x}, \mathbf{u}), \qquad \delta'^2_{z_i} = \beta'_{i,2}(\mathbf{x}, \mathbf{u}). \tag{11}$$

Therefore, we can arrive at a simple objective:

$$\max \mathbb{E}_{q(\mathbf{z}|\mathbf{x}, \mathbf{u})}(p(\mathbf{x}|\mathbf{z}, \mathbf{u})) - D_{KL}(q(\mathbf{z}|\mathbf{x}, \mathbf{u})||p(\mathbf{z}|\mathbf{u})), \tag{12}$$

where $D_{KL}$ denotes Kullback–Leibler divergence. Note that the proposed SuaVE is different from hierarchical VAE models (Kingma et al., 2016; Sønderby et al., 2016; Maaløe et al., 2019; Vahdat & Kautz, 2020) in that the former has rigorous theoretical justification for identifiability, while the latter has no such supports without further assumptions. Besides, the theoretical justification implies **the consistency of estimation** for the proposed SuaVE as follows:

**Theorem 5.1** *Assume the following holds:*

*(i) The variational distributions Eq. 10 contains the ture posterior distribution $p_{\mathbf{f}, \boldsymbol{\lambda}, \boldsymbol{\beta}}(\mathbf{z}|\mathbf{x}, \mathbf{u})$.*

*(ii) We maximize the objective function: $\max \mathbb{E}_{q(\mathbf{z}|\mathbf{x}, \mathbf{u})}(p(\mathbf{x}|\mathbf{z}, \mathbf{u})) - D_{KL}(q(\mathbf{z}|\mathbf{x}, \mathbf{u})||p(\mathbf{z}|\mathbf{u}))$.*

*then in the limit of infinite data, the proposed SuaVE learns the true parameters $\boldsymbol{\theta} = (\mathbf{f}, \boldsymbol{\lambda}, \boldsymbol{\beta})$ up to $\mathbf{z} = \mathbf{P}\hat{\mathbf{z}} + \mathbf{c}$, where $\mathbf{P}$ denotes the permutation matrix with scaling, $\mathbf{c}$ denotes a constant vector.*

## 6 EXPERIMENTS

**Synthetic Data**    We first conduct experiments on synthetic data, generated by the following process: we divide the latent noise variables into $M$ segments, where each segment corresponds to one conditional variable $\mathbf{u}$ as the segment label. Within each segment, we first sample the mean $\beta_{i,1}$ and variance $\beta_{i,2}$ from uniform distributions $[-2, 2]$ and $[0.01, 3]$, respectively. We then sample the weights $\boldsymbol{\lambda}_{i,j}$ from uniform distributions $[0.1, 2]$. Then for each segment, we generate the latent causal samples according to the generative model in Eq.3. Finally, we obtain the observed data samples $\mathbf{x}$ by an invertible nonlinear mapping. More details can be found in Appendix.

**Comparison**    We compare the proposed method with identifiable VAE (iVAE) Khemakhem et al. (2020), $\beta$-VAE Higgins et al. (2017), CausalVAEYang et al. (2021), and vanilla VAE Kingma & Welling (2013b). Among them, iVAE has been proven to be identifiable so that it is able to learn the true independent noise variables with certain assumptions. While $\beta$-VAE has no theoretical support, it has been widely used in various disentanglement tasks. Note that both methods assume that the latent variables are independent, and thus they cannot model the relationships among latent variables. To make a fair comparison in the unsupervised setting, we implement an unsupervised version of CausalVAE, which is not identifiable.

**Performance metric**    Since the proposed method can recover the latent causal variables up to trivial permutation and linear scaling, we compute the mean of the Pearson correlation coefficient (MPC) to evaluate the performance of our proposed method. The Pearson correlation coefficient is a measure of linear correlation between the true latent causal variables and the recovered latent causal variables. Note that the Pearson coefficient is suitable for iVAE, since it has been shown that iVAE can also recover the latent noise variables up to linear scaling under the setting where mean and variance of the latent noise variables are changed by $\mathbf{u}$ (Sorrenson et al., 2020). To remove the permutation effect, following Khemakhem et al. (2020), we first calculate all pairs of correlation and then solve a linear sum assignment problem to obtain the final results. A high correlation coefficient means that we successfully identified the true parameters and recovered the true variables, up to component-wise linear transformations.

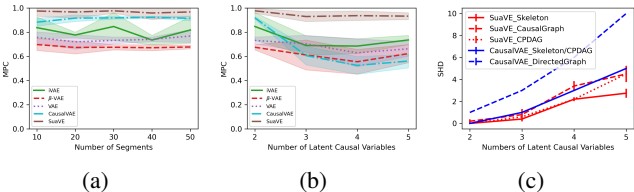

(a)          (b)          (c)

Figure 5: Performances of the proposed SuaVE in comparison to iVAE, $\beta$-VAE, VAE and CausalVAE in recovering the latent causal variables on synthetic data with different numbers of segments (a), and with different numbers of the latent causal variables (b). The proposed SuaVE obtain the best MPC, compared with the others. (c) The SHD obtained by the proposed SuaVE and CausalVAE.

**Results**    We compared the performance of the proposed SuaVE to some variants of VAE mentioned above. We used the same network architecture for encoder and decoder parts in all these models. In particular, we add a sub-network to model the conditional prior in both iVAE and the proposed SuaVE. We further assign an linear SCM sub-network to model the relations among latent causal variables in the proposed SuaVE. We trained these 5 models on the dataset described above, with different numbers of segments and different numbers of latent causal variables. For each method, we use 5 different random seeds for data sampling. Figure 5 (a) shows the performance on two latent causal variables with different numbers of segment. The proposed SuaVE obtains the score 0.96 approximately for all the different numbers of segment. In contrast, $\beta$-VAE, VAE and CausalVAE fail to achieve a good estimation of the true latent variables, since they are not identifiable. iVAE obtains unsatisfactory results, since its theory only holds for i.i.d. latent variables. Figure 5 (b) shows the performance in recovering latent causal variables on synthetic data with different numbers of the latent causal variables. Figure 5 (c) depicts the structural Hamming distance (SHD) of the recovered skeletons, causal graphs and completed partial directed acyclic graphs (CPDAG) by the proposed SuaVE and CausalVAE. Since CausalVAE obtains fully connected graphs, the SHD of the recovered CPDAG is the same as one of the recovered skeletons.

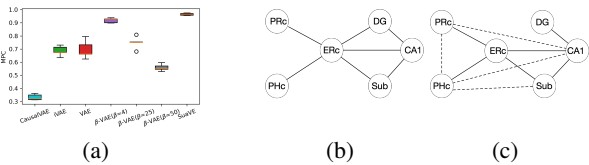

(a)                          (b)                          (c)

Figure 6: Performance on fMRI Data. (a) the performance of the proposed SuaVE in comparison to iVAE, $\beta$-VAE, VAE and CausalVAE in recovering the latent six signals. (b) The skeleton of the anatomical connections given in Bird & Burgess (2008). (c) The recovered skeleton by the proposed SuaVE, where the dashed lines indicate errors.

**fMRI Data**   Following Ghassami et al. (2018), we applied the proposed method to fMRI hippocampus dataset (Laumann & Poldrack, 2015), which contains signals from six separate brain regions: perirhinal cortex (PRC), parahippocampal cortex (PHC), entorhinal cortex (ERC), subiculum (Sub), CA1, and CA3/Dentate Gyrus (CA3/DA) in the resting states on the same person in 84 successive days. Each day is considered as different $\mathbf{u}$, thus $\mathbf{u}$ is a 84-dimensional vector. Since we are interested in recovering latent causal variables, we treat the six signals as latent causal variables by applying a random nonlinear mapping on them to obtain observed data. We then apply various methods on the observed data to recover the six signals. Figure 6 (a) shows the performance of the proposed SuaVE in comparison to iVAE, $\beta$-VAE, VAE and CausalVAE in recovering the latent six signals. $\beta$-VAE aims to recover independent latent variables, and it obtains an interesting result: enforcing independence (*e.g.* $\beta = 25, 50$) leads to worse MPC, and relaxing it (though contracting to its own independence assumption) improves the result (*e.g.* $\beta = 4$). This is because the latent variables given the time index are not independent in this dataset.

We further verify the ability of the proposed method to recover causal skeleton. We use the anatomical connections (Bird & Burgess, 2008) as a reference as shown in Figure 6 (b), since causal connections among the six regions should not exist if there is no anatomical connection. After recovering the latent causal variables, we estimate the causal skeleton by a threshold (0.1) on the learned $\lambda_{i,j}$ to remove some edges, and obtain the final causal skeleton. Figure 6 (c) shows an example of the estimated causal skeleton by the proposed SuaVE. Averaging over 5 different random seeds for the used nonlinear mapping from latent space to observed space, structural Hamming distance (SHD) obtained by the proposed SuaVE is **5.0 ± 0.28**. In contrast, iVAE, $\beta$-VAE and VAE assume latent variables to be independent, and thus can not obtain the causal skeleton. We further analyse the result of $\beta$-VAE with $\beta = 4$ by using the PC algorithm (Spirtes et al., 2001) to discovery skeleton on the recovered latent variables, and obtain **SHD = 5.8 ± 0.91**. This means that: 1) $\beta$-VAE with $\beta = 4$ does not ensure the strong independence among the recover latent variables as it is expected, 2) the proposed SuaVE is an effective one-step method to discovery the skeleton. The unsupervised version of CausalVAE is not identifiable, and can also not ensure the relations among the recover variables to be causal skeleton in principle. In experiments we found that CausalVAE always obtains fully-connected graphs for the 5 different random seeds, for which SHD is **9.0 ± 0**.

## 7 CONCLUSION

Identifying latent causal representations is known to be generally impossible without certain assumptions. Motivated by recent progress in the identifiability result of nonlinear ICA, we analyse three intrinsic indeterminacies in latent space, *e.g.*, transitivity, permutation indeterminacy and scaling indeterminacy, which provide deep insights for identifying latent causal representations. To address the unidentifiable issue due to transitivity, we explore a new line of research in the identifiability of latent causal representations, that of allowing the causal influences among the latent causal variables to change by an additionally observed variable. We show that latent causal representations, modelled by weight-variant linear Gaussian models, can be identified up to permutation and scaling, with mild assumptions. We then propose a novel method to consistently learn latent causal representations. Experiments on synthetic and fMRI data demonstrate the identifiability and consistency results and the efficacy of the proposed method. Future work may include developing effective methods for learning latent causal structures and exploring nonlinear causal representations.

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

# A  APPENDIX

## A.1  THE PROOF OF THEOREM 4.1

The proof of Theorem 4.1 is done in three steps. The steps I is the same as nonlinear ICA Khemakhem et al. (2020), which builds a connection between any two different solutions by using a simple convolutional trick made possible by assumption (i). The step II consists of building a linear connection between any two different solutions by assumptions (iv). This step is different with nonlinear ICA Khemakhem et al. (2020) in that we employ the assumptions of multivariate Gaussian as Eq. 6, which can model the linear causal relations among the latent causal variables, while nonlinear ICA Khemakhem et al. (2020) assumes independent noise variables. The step III shows that we can recover latent causal variables up to a linear transformation by the identity theorem for polynomials and assumption (iii). The step IV shows that the linear transformation can be reduced to permutation transformation by assumption (iv).

**Step I:** Suppose we have two sets of parameters $\theta = (\mathbf{f}, \mathbf{T}, \boldsymbol{\lambda}, \boldsymbol{\beta})$ and $\hat{\theta} = (\hat{\mathbf{f}}, \hat{\mathbf{T}}, \hat{\boldsymbol{\lambda}}, \hat{\boldsymbol{\beta}})$ such that $p_{(\mathbf{f}, \mathbf{T}, \boldsymbol{\lambda}, \boldsymbol{\beta})}(\mathbf{x}|\mathbf{u}) = p_{(\hat{\mathbf{f}}, \hat{\mathbf{T}}, \hat{\boldsymbol{\lambda}}, \hat{\boldsymbol{\beta}})}(\mathbf{x}|\mathbf{u})$ for all pairs $(\mathbf{x}, \mathbf{u})$. Then:

$$\int p_{(\mathbf{T}, \boldsymbol{\lambda}, \boldsymbol{\beta})}(\mathbf{z}|\mathbf{u}) p_{(\mathbf{f})}(\mathbf{x}|\mathbf{z}) \, \mathrm{d}\mathbf{z} = \int p_{(\hat{\mathbf{T}}, \hat{\boldsymbol{\lambda}}, \hat{\boldsymbol{\beta}})}(\mathbf{z}|\mathbf{u}) p_{(\hat{\mathbf{f}})}(\mathbf{x}|\mathbf{z}) \, \mathrm{d}\mathbf{z} \tag{13}$$

$$\Rightarrow \int p_{(\mathbf{T}, \boldsymbol{\lambda}, \boldsymbol{\beta})}(\mathbf{z}|\mathbf{u}) p_\varepsilon(\mathbf{x} - \mathbf{f}(\mathbf{z})) \, \mathrm{d}\mathbf{z} = \int p_{(\hat{\mathbf{T}}, \hat{\boldsymbol{\lambda}}, \hat{\boldsymbol{\beta}})}(\mathbf{z}|\mathbf{u}) p_\varepsilon(\mathbf{x} - \hat{\mathbf{f}}(\mathbf{z})) \, \mathrm{d}\mathbf{z} \tag{14}$$

$$\Rightarrow \int p_{(\mathbf{T}, \boldsymbol{\lambda}, \boldsymbol{\beta})}(\mathbf{f}^{-1}(\bar{\mathbf{x}})|\mathbf{u}) vol J_{\mathbf{f}^{-1}}(\bar{\mathbf{x}}) p_\varepsilon(\mathbf{x} - \bar{\mathbf{x}}) \, \mathrm{d}\bar{\mathbf{x}} = \int p_{(\hat{\mathbf{T}}, \hat{\boldsymbol{\lambda}}, \hat{\boldsymbol{\beta}})}(\hat{\mathbf{f}}^{-1}(\bar{\mathbf{x}}|\mathbf{u})) vol \hat{J}_{\hat{\mathbf{f}}^{-1}}(\bar{\mathbf{x}}) p_\varepsilon(\mathbf{x} - \bar{\mathbf{x}}) \, \mathrm{d}\bar{\mathbf{x}} \tag{15}$$

$$\Rightarrow \int \hat{p}_{(\mathbf{T}, \boldsymbol{\lambda}, \boldsymbol{\beta}, \mathbf{f}, \mathbf{u})}(\bar{\mathbf{x}}) p_\varepsilon(\mathbf{x} - \bar{\mathbf{x}}) \, \mathrm{d}\bar{\mathbf{x}} = \int \hat{p}_{(\hat{\mathbf{T}}, \hat{\boldsymbol{\lambda}}, \hat{\boldsymbol{\beta}}, \hat{\mathbf{f}}, \mathbf{u})}(\bar{\mathbf{x}}) p_\varepsilon(\mathbf{x} - \bar{\mathbf{x}}) \, \mathrm{d}\bar{\mathbf{x}} \tag{16}$$

$$\Rightarrow (\hat{p}_{(\mathbf{T}, \boldsymbol{\lambda}, \boldsymbol{\beta}, \mathbf{f}, \mathbf{u})} * p_\varepsilon)(\mathbf{x}) = (\hat{p}_{(\hat{\mathbf{T}}, \hat{\boldsymbol{\lambda}}, \hat{\boldsymbol{\beta}}, \hat{\mathbf{f}}, \mathbf{u})} * p_\varepsilon)(\mathbf{x}) \tag{17}$$

$$\Rightarrow F[\hat{p}_{(\mathbf{T}, \boldsymbol{\lambda}, \boldsymbol{\beta}, \mathbf{f}, \mathbf{u})}](\omega) \varphi_\varepsilon(\omega) = F[\hat{p}_{(\hat{\mathbf{T}}, \hat{\boldsymbol{\lambda}}, \hat{\boldsymbol{\beta}}, \hat{\mathbf{f}}, \mathbf{u})}](\omega) \varphi_\varepsilon(\omega) \tag{18}$$

$$\Rightarrow F[\hat{p}_{(\mathbf{T}, \boldsymbol{\lambda}, \boldsymbol{\beta}, \mathbf{f}, \mathbf{u})}](\omega) = F[\hat{p}_{(\hat{\mathbf{T}}, \hat{\boldsymbol{\lambda}}, \hat{\boldsymbol{\beta}}, \hat{\mathbf{f}}, \mathbf{u})}](\omega) \tag{19}$$

$$\Rightarrow \hat{p}_{(\mathbf{T}, \boldsymbol{\lambda}, \boldsymbol{\beta}, \mathbf{f}, \mathbf{u})}(\mathbf{x}) = \hat{p}_{(\hat{\mathbf{T}}, \hat{\boldsymbol{\lambda}}, \hat{\boldsymbol{\beta}}, \hat{\mathbf{f}}, \mathbf{u})}(\mathbf{x}) \tag{20}$$

where:

- in Eq. 15, $J$ denotes the Jacobian, and we introduce here the volume of a matrix denoted $vol J$ as the product of the singular values of $J$. When $J$ is full column rank, $vol J = \sqrt{\det J^T J}$, and when $J$ is invertible, $vol J = |\det J|$. We made the change of variable $\bar{\mathbf{x}} = \mathbf{f}(\mathbf{z})$ on the left hand side, and $\bar{\mathbf{x}} = \hat{\mathbf{f}}(\mathbf{z})$ on the right hand side.
- in Eq. 16, we introduced:

$$\hat{p}_{(\mathbf{T}, \boldsymbol{\lambda}, \boldsymbol{\beta}, \mathbf{f}, \mathbf{u})}(\mathbf{x}) = p_{(\mathbf{T}, \boldsymbol{\lambda}, \boldsymbol{\beta})}(\mathbf{f}^{-1}(\mathbf{x})|\mathbf{u}) vol J_{\mathbf{f}^{-1}}(\mathbf{x}) \mathbb{1}_{\mathcal{X}}(\mathbf{x}), \tag{21}$$

  on the left hand side, and similarly on the right hand side.
- in Eq. 17, we used $*$ for the convolution operator.
- in Eq. 18, we used $F[.]$ to designate the Fourier transform.
- in Eq. 19, we dropped $\varphi_\varepsilon$ from both sides as it is non-zero almost everywhere (by assumption (i)).

**Step II** By taking the logarithm on both sides of Eq. 20 and replacing $p_{(\mathbf{T}, \boldsymbol{\lambda}, \boldsymbol{\beta})}$ by its expression from Eq. 6, we have:

$$\log vol J_{\mathbf{f}^{-1}}(\mathbf{x}) + \mathbf{T}^T(\mathbf{f}^{-1}(\mathbf{x})) \boldsymbol{\eta}(\mathbf{u}) - \log Z(\boldsymbol{\lambda}, \boldsymbol{\beta}, \mathbf{u}) = \tag{22}$$

$$\log vol J_{\hat{\mathbf{f}}^{-1}}(\mathbf{x}) + \hat{\mathbf{T}}^T(\hat{\mathbf{f}}^{-1}(\mathbf{x})) \hat{\boldsymbol{\eta}}(\mathbf{u}) - \log \hat{Z}(\hat{\boldsymbol{\lambda}}, \hat{\boldsymbol{\beta}}, \mathbf{u}). \tag{23}$$

Let $\mathbf{u}_0, \mathbf{u}_1, ..., \mathbf{u}_{k+1}$ be the points provided by assumption (iii) of the Theorem. Similar to nonlinear ICA Khemakhem et al. (2020), we define $\bar{\boldsymbol{\eta}}(\mathbf{u}) = \boldsymbol{\eta}(\mathbf{u}) - \boldsymbol{\eta}(\mathbf{u}_0)$. We plug each of those $\mathbf{u}_l, l = 1, ..., k$ in Eq. 23 to obtain $k + 1$ equations. We subtract the first equation for $\mathbf{u}_0$ from the remaining $k$ equations. For example, for $\mathbf{u}_0$ and $\mathbf{u}_k$ we have two equations:

$$\log \text{vol} J_{\mathbf{f}^{-1}}(\mathbf{x}) + \mathbf{T}^T(\mathbf{f}^{-1}(\mathbf{x}))\boldsymbol{\eta}(\mathbf{u}_0) - \log Z(\boldsymbol{\lambda}, \boldsymbol{\beta}, \mathbf{u}_0) =$$
$$\log \text{vol} J_{\hat{\mathbf{f}}^{-1}}(\mathbf{x}) + \hat{\mathbf{T}}^T(\hat{\mathbf{f}}^{-1}(\mathbf{x}))\hat{\boldsymbol{\eta}}(\mathbf{u}_0) - \log \hat{Z}(\hat{\boldsymbol{\lambda}}, \hat{\boldsymbol{\beta}}, \mathbf{u}_0),$$
(24)

and

$$\log \text{vol} J_{\mathbf{f}^{-1}}(\mathbf{x}) + \mathbf{T}^T(\mathbf{f}^{-1}(\mathbf{x}))\boldsymbol{\eta}(\mathbf{u}_k) - \log Z(\boldsymbol{\lambda}, \boldsymbol{\beta}, \mathbf{u}_k) =$$
$$\log \text{vol} J_{\hat{\mathbf{f}}^{-1}}(\mathbf{x}) + \hat{\mathbf{T}}^T(\hat{\mathbf{f}}^{-1}(\mathbf{x}))\hat{\boldsymbol{\eta}}(\mathbf{u}_k) - \log \hat{Z}(\hat{\boldsymbol{\lambda}}, \hat{\boldsymbol{\beta}}, \mathbf{u}_k),$$
(25)

Using Eq. 25 subtracts Eq. 24, removing the terms that do not include $\mathbf{u}$, we have:

$$\mathbf{T}^T(\mathbf{f}^{-1}(\mathbf{x}))\boldsymbol{\eta}(\mathbf{u}_k) - \log Z(\boldsymbol{\lambda}, \boldsymbol{\beta}, \mathbf{u}_k) - \mathbf{T}^T(\mathbf{f}^{-1}(\mathbf{x}))\boldsymbol{\eta}(\mathbf{u}_0) + \log Z(\boldsymbol{\lambda}, \boldsymbol{\beta}, \mathbf{u}_0)$$
$$= \hat{\mathbf{T}}^T(\hat{\mathbf{f}}^{-1}(\mathbf{x}))\hat{\boldsymbol{\eta}}(\mathbf{u}_k) - \log \hat{Z}(\hat{\boldsymbol{\lambda}}, \hat{\boldsymbol{\beta}}, \mathbf{u}_k) - \hat{\mathbf{T}}^T(\hat{\mathbf{f}}^{-1}(\mathbf{x}))\hat{\boldsymbol{\eta}}(\mathbf{u}_0) + \log \hat{Z}(\hat{\boldsymbol{\lambda}}, \hat{\boldsymbol{\beta}}, \mathbf{u}_0).$$
(26)

As a result, we arrive at:

$$\langle \mathbf{T}(\mathbf{f}^{-1}(\mathbf{x})), \bar{\boldsymbol{\eta}}(\mathbf{u}_l) \rangle + \log \frac{Z(\boldsymbol{\lambda}, \boldsymbol{\beta}, \mathbf{u}_0)}{Z(\boldsymbol{\lambda}, \boldsymbol{\beta}, \mathbf{u}_l)} = \langle \hat{\mathbf{T}}(\hat{\mathbf{f}}^{-1}(\mathbf{x})), \bar{\hat{\boldsymbol{\eta}}}(\mathbf{u}_l) \rangle + \log \frac{\hat{Z}(\hat{\boldsymbol{\lambda}}, \hat{\boldsymbol{\beta}}, \mathbf{u}_0)}{\hat{Z}(\hat{\boldsymbol{\lambda}}, \hat{\boldsymbol{\beta}}, \mathbf{u}_l)}. \quad (27)$$

Let $\mathbf{L}$ be the matrix defined in Eq. 7 in assumption (iii), and $\hat{\mathbf{L}}$ similarly ($\hat{\mathbf{L}}$ is not necessarily invertible). Define $b_l = \log \frac{\hat{Z}(\hat{\boldsymbol{\lambda}}, \hat{\boldsymbol{\beta}}, \mathbf{u}_0) Z(\boldsymbol{\lambda}, \boldsymbol{\beta}, \mathbf{u}_l)}{Z(\boldsymbol{\lambda}, \boldsymbol{\beta}, \mathbf{u}_0) \hat{Z}(\hat{\boldsymbol{\lambda}}, \hat{\boldsymbol{\beta}}, \mathbf{u}_l)}$ and $\mathbf{b}$ the vector for $b_l$. Expressing Eq. 27 for all points $\mathbf{u}_l$ in matrix form, we get:

$$\mathbf{L}^T \mathbf{T}(\mathbf{f}^{-1}(\mathbf{x})) = \hat{\mathbf{L}}^T \hat{\mathbf{T}}(\hat{\mathbf{f}}^{-1}(\mathbf{x})) + \mathbf{b}. \quad (28)$$

We multiply both sides of Eq. 28 by the inverse of $\mathbf{L}^T$ to find:

$$\mathbf{T}(\mathbf{f}^{-1}(\mathbf{x})) = \mathbf{A}\hat{\mathbf{T}}(\hat{\mathbf{f}}^{-1}(\mathbf{x})) + \mathbf{c}, \quad (29)$$

where $\mathbf{A} = (\mathbf{L}^T)^{-1}\hat{\mathbf{L}}^T$ and $\mathbf{c} = (\mathbf{L}^T)^{-1}\mathbf{b}$.

**Step III** As mentioned in Eq. 6, the sufficient statistic $\mathbf{T}(\mathbf{z}) = [\mathbf{z}; \text{vec}(\mathbf{z}\mathbf{z}^T)]$. In this case, the relationship Eq. 29 becomes:

$$\begin{pmatrix} \mathbf{z} \\ \mathbf{z}^2 \\ \mathbf{z}_{i,j} \end{pmatrix} = \mathbf{A} \begin{pmatrix} \hat{\mathbf{z}} \\ \hat{\mathbf{z}}^2 \\ \hat{\mathbf{z}}_{i,j} \end{pmatrix} + \mathbf{c}, \quad (30)$$

where $\mathbf{z}$ denotes $[z_1, ..., z_i]$, $\mathbf{z}^2$ denotes $[z_1^2, ..., z_i^2]$, $\mathbf{z}_{i,j}$ denotes the vector whose elements are $z_i z_j$ for all $i \neq j$, $\mathbf{A}$ in block matrix can be rewritten as:

$$\mathbf{A} = \begin{pmatrix} \mathbf{A}^{(1)} & \mathbf{A}^{(2)} & \mathbf{A}^{(3)} \\ \mathbf{A}^{(4)} & \mathbf{A}^{(5)} & \mathbf{A}^{(6)} \\ \mathbf{A}^{(7)} & \mathbf{A}^{(8)} & \mathbf{A}^{(9)} \end{pmatrix} \quad (31)$$

and $\mathbf{c}$ as:

$$\mathbf{c} = \begin{pmatrix} \mathbf{c}^{(1)} \\ \mathbf{c}^{(2)} \\ \mathbf{c}^{(3)} \end{pmatrix}. \quad (32)$$

Then, we have:

$$\mathbf{z} = \mathbf{A}^{(1)}\hat{\mathbf{z}} + \mathbf{A}^{(2)}\hat{\mathbf{z}}^2 + \mathbf{A}^{(3)}\hat{\mathbf{z}}_{i,j} + \mathbf{c}^{(1)}, \quad (33)$$

$$\mathbf{z}^2 = \mathbf{A}^{(4)}\hat{\mathbf{z}} + \mathbf{A}^{(5)}\hat{\mathbf{z}}^2 + \mathbf{A}^{(6)}\hat{\mathbf{z}}_{i,j} + \mathbf{c}^{(2)}. \quad (34)$$

So we can write for each $z_i$:

$$z_i = \sum_j (A_{i,j}^{(1)}\hat{z}_j) + \sum_j (A_{i,j}^{(2)}\hat{z}_j^2) + \sum_{j \neq i} (A_{i,j}^{(3)}\hat{z}_i\hat{z}_j) + c_i^{(1)}, \quad (35)$$

$$z_i^2 = \sum_j (A_{i,j}^{(4)}\hat{z}_j) + \sum_j (A_{i,j}^{(5)}\hat{z}_j^2) + \sum_{j \neq i} (A_{i,j}^{(6)}\hat{z}_i\hat{z}_j) + c_i^{(2)}. \quad (36)$$

Squaring Eq. 35, we have:

$$z_i^2 = \underbrace{(\sum_j (A_{i,j}^{(2)} \hat{z}_j^2))^2}_{(a)} + \underbrace{(\sum_j (A_{i,j}^{(1)} \hat{z}_j))^2}_{(b)} + \underbrace{(\sum_{j \neq i} (A_{i,j}^{(3)} \hat{z}_i \hat{z}_j))^2}_{(c)} + (c_i^{(1)})^2 + ..... \tag{37}$$

It is notable that Eq. 36 and Eq. 37 are derived from Eq. 29 which holds for arbitrary $\mathbf{x}$, then by the assumption (ii) in Theorem 4.1 that $\mathbf{f}$ is a bijective mapping from $\mathbf{z}$ to $\mathbf{x}$, and thus Eq. 36 and Eq. 37 must holds for $\mathbf{z}$ everywhere. Then by the fact that the right sides of the Eq. 36 and Eq. 37 are both polynomials with finite degree, we have each coefficients of the two polynomials must be equal. In more detail, for the term (a) in Eq. 37:

$$(\sum_j (A_{i,j}^{(2)} \hat{z}_j^2))^2 = \sum_j (A_{i,j}^{(2)})^2 \hat{z}_j^4 + \sum_{j \neq j'} (2 A_{i,j}^{(2)} A_{i,j'}^{(2)} \hat{z}_j^2 \hat{z}_{j'}^2). \tag{38}$$

Compared with Eq. 36, since there is no term $\hat{z}_j^4$ in Eq. 36, we must have that:

$$\mathbf{A}^{(2)} = 0. \tag{39}$$

For the term (c) in Eq. 37:

$$(\sum_{j \neq i} (A_{i,j}^{(3)} \hat{z}_i \hat{z}_j))^2 = \sum_{j \neq i} (A_{i,j}^{(3)})^2 (\hat{z}_i \hat{z}_j)^2 + ... \tag{40}$$

Compared with Eq. 36, since there is no term $(\hat{z}_i \hat{z}_j)^2$ in Eq. 36, we must have that:

$$\mathbf{A}^{(3)} = 0 \tag{41}$$

As a result, Eq. 33 becomes:

$$\mathbf{z} = \mathbf{A}^{(1)} \hat{\mathbf{z}} + \mathbf{c}^{(1)}. \tag{42}$$

The above equation indicates the latent causal variables can be recovered up to linear transformation.

**Step IV** We then show that the linear transformation matrix $\mathbf{A}^{(1)}$ in Eq. 42 must be a permutation matrix. Since our assumptions in Theorem 4.1 includes the assumptions for identifying the Gaussian noise variables $n_i$, the Gaussian noise variables $n_i$ can be recovered up to linear scaling and permutation according to Sorrenson et al. (2020):

$$\mathbf{n} = \mathbf{P}\hat{\mathbf{n}} + \mathbf{c}_n, \tag{43}$$

where $\mathbf{P}$ denote the permutation matrix. With Eq. 43, the Eq. 42 can be rewritten as follows:

$$\mathbf{B}(\mathbf{P}\hat{\mathbf{n}} + \mathbf{c}_n) = \mathbf{A}^{(1)}(\hat{\mathbf{B}}\hat{\mathbf{n}}) + \mathbf{c}^{(1)}. \tag{44}$$

$$\Rightarrow (\mathbf{BP} - \mathbf{A}^{(1)}\hat{\mathbf{B}})\hat{\mathbf{n}} = \mathbf{A}^{(1)}\hat{\mathbf{B}}\mathbf{c}_n + \mathbf{c}^{(1)} - \mathbf{B}\mathbf{c}_n, \tag{45}$$

where $\mathbf{B}$ and $\hat{\mathbf{B}}$ denote lower triangular matrix, for which the main diagonal are 1, the others depend on $\lambda_{i,j}(\mathbf{u})$ and $\hat{\lambda}_{i,j}(\mathbf{u})$, respectively. By differentiating Eq. 45 with respect to $\hat{\mathbf{n}}$, we have:

$$(\mathbf{BP} - \mathbf{A}^{(1)}\hat{\mathbf{B}})\mathbf{I} = \mathbf{0}, \tag{46}$$

where $\mathbf{I}$ denote the identity matrix, which implies that:

$$\mathbf{BP} = \mathbf{A}^{(1)}\hat{\mathbf{B}}. \tag{47}$$

Then $\mathbf{A}^{(1)}$ must be an identity matrix. To understand this point, we give a 2-dimensional case as follows, which can be easily extended to n-dimensional case. Without loss of generality, we assume $\mathbf{P}$ to be diagonal with elements $s_{1,1}, s_{2,2}$.

$$\begin{pmatrix} s_{1,1} & 0 \\ s_{1,1}\lambda_{1,2}(\mathbf{u}) & s_{2,2} \end{pmatrix} = \begin{pmatrix} A_{1,1}^{(1)} & A_{1,2}^{(1)} \\ A_{2,1}^{(1)} & A_{2,2}^{(1)} \end{pmatrix} \begin{pmatrix} 1 & 0 \\ \hat{\lambda}_{1,2}(\mathbf{u}) & 1 \end{pmatrix} = \begin{pmatrix} A_{1,1}^{(1)} + A_{1,2}^{(1)}\hat{\lambda}_{1,2}(\mathbf{u}) & A_{1,2}^{(1)} \\ A_{2,1}^{(1)} + A_{2,2}^{(1)}\hat{\lambda}_{1,2}(\mathbf{u}) & A_{2,2}^{(1)} \end{pmatrix}. \tag{48}$$

Then we have: $A_{1,2}^{(1)} = 0$, $A_{1,1}^{(1)} = s_{1,1}$, $A_{2,2}^{(1)} = s_{2,2}$, and $A_{2,1}^{(1)} = 0$ (by our assumption (iv) $\lambda_{i,j}(\mathbf{0}) = 0$). The proof of other case for $\mathbf{P}$ is similar to the above procedure.

A.2    UNDERSTANDING ASSUMPTIONS IN THEOREM 4.1

**Gaussian assumption on the latent noise variables**    Our first assumption is implicitly enforcing Gaussian distribution on the latent noise variables. Note that the assumptions of nonlinear ICA (Khemakhem et al., 2020) on the noise could be broad exponential family distribution, *e.g.*, Gaussian distributions, Laplace, Gamma distribution and so on. This work consider Gaussian distribution, mainly because it can be straightforwardly implemented for the re-parameterization trick in VAE. 2) it is convenient for simplifying proof process (As shown in Step III in the proof.). It is worthwhile and promising to extend the Gaussian assumption to exponential family for future work.

**Linear model assumption for the latent causal variables**    As the first work to discuss the relation between the change of causal influences and identifiability pf latent causal model, this work simply consider linear models for the latent causal variables, because it can be directly parameterized, which helps us to analyse the challenge of identifiability and how to handle it. In addition, together with Gaussian noise, we arrive the same probabilistic model as linear Gaussian Bayesian networks, which is a well-studied model as mentioned in Eq. 5. We expect that the proposed weights-variant linear models can motivate more general functional class in the future work, *e.g.*, nonlinear additive noise models.

**Changes of weight and Assumption (iv)**    We allow changes of weight (*e.g.*, causal influences) among latent causal variables across $\mathbf{u}$ to handle the transitivity problem. We argue that this is not necessary condition for identifiability (Existing methods, including sparse graph structure and temporal information, could be regarded as feasible ways to handle the transitivity.), but it is a sufficient condition to handle the transitivity, and provide a new research line for causal representation learning. Assumption (iv) is to handle a special case as follows: $\lambda_{i,j}(\mathbf{u}) = \lambda'_{i,j}\mathbf{u} + b$ where $b$ is a non-zero constant. For example, if $z_2 = (\lambda'_{1,2}(\mathbf{u}) + b)z_1 + n_2$, then the term $bz_1$ is still unchanged across all $\mathbf{u}$, and thus can still be 'absorbed' by the nonlinear mapping from latent variables $\mathbf{z}$ to the observed variable $\mathbf{x}$ due to the transitivity, which results in non-identifiability. Note that assumption (iv) is sufficient to handle the special case, but it may not be necessary. There may exist possible assumption to relax the assumption (iv).

**Assumptions (i)-(iii)**    All three assumptions are motivated by the nonlinear ICA literature (Khemakhem et al., 2020), which is to provide guarantee that we can recover latent causal variables $z_i$ up to a linear transformation. Note that three assumptions also provide a support for recovering the latent noise variables up to linear scaling and permutation.

A.3    THE PROOF OF COROLLARY 4.2

Theorem 4.1 has shown that the latent causal variables $\mathbf{z}$ can be identified up to trivial permutation and linear scaling. Hence, the identifiability of the causal structure in latent space can be reduced to the identifiability of the causal structure in observed space. Moreover, we need to show that the linear scaling does not affect theoretical identifiability of the causal structure.

For the identifiability of the causal structure in observed space from heterogeneous data, fortunately, we can leverage the results from Huang* et al. (2020). Corollary 4.2 relies on the Markov condition and faithfulness assumption and the assumption that the latent change factor (i.e., causal strength in the linear case) can be represented as a function of the domain index $\mathbf{u}$. Hence, it relies on the same assumptions as that in Huang* et al. (2020).

It has been shown in Huang* et al. (2020) that if the joint distribution over $\mathbf{z}$ and $\mathbf{u}$ ($\mathbf{z}$ and $\mathbf{u}$ are observed variables here) are Markov and faithful to the augmented graph, then the causal structure over $\mathbf{z} \cup \mathbf{u}$ can be identified up to the Markov equivalence class, by making use of the conditional independence relationships.

Next, we show that the Markov equivalence class over $\mathbf{z}$ is also identifiable. Denote by $M$ the Markov equivalence class over $\mathbf{z} \cup \mathbf{u}$, and by $M_z$ the Markov equivalence class over $\mathbf{z}$. Then after removing variable $\mathbf{u}$ in $M$ and its edges, the resulting graph (denoted by $M'_z$) is the same as $M_z$. This is because of the following reasons. First, it is obvious that $M'_z$ and $M_z$ have the same skeleton. Second, in

this paper, $\mathbf{u}$ has an edge to every $z_i$ when considering $\mathbf{n}$ as latent noise variables, because all causal strength and noise distributions change with $\mathbf{u}$. Hence, there is no v-structure over $\mathbf{u}$, $z_i$, and $z_j$, so it is not possible to have more oriented edges in $M_z'$. Therefore, $M_z'$ and $M_z$ have the same skeleton and the same directions.

Moreover, the conditional independence relationships will not be affected by linear scaling of the variables, so the conditional independence relationships still hold in the identified latent variables. Furthermore, for linear-Gaussian models, independence equivalence is the same as distributional equivalence. This is because of the following reasons. First, since we are concerned with linear Gaussian models over the latent variables, the identifiability up to equivalence class also holds for score-based methods that use the likelihood of data as objective functions. This is because independence equivalence, i.e., two DAGs have the same conditional independence relations, is the same as distributional equivalence for linear-Gaussian models, i.e., two Bayesian networks corresponding to the two DAGs can define the same probability distribution. That is to say, score-based methods also find the structure based on independence relations implicitly. Because the scaling does not change the independence relations, it will also not affect the identifiability of the graph structure.

Therefore, the causal structure among latent variables $\mathbf{z}$ can be identified up to the Markov equivalence class.

### A.4 THE PROOF OF THEOREM 5.1

The proof of Theorem 5.1 is similar to the proof of Theorem 4 in non-linear ICA Khemakhem et al. (2020). If the variational posterior $q(\mathbf{z}|\mathbf{x}, \mathbf{u})$ is large enough to include the true posterior $p(\mathbf{z}|\mathbf{x}, \mathbf{u})$, then by optimizing the loss, the KL term will be zero, and the loss will be equal to the log-likelihood. Since our identifiability is guaranteed up to permutation and scaling, the consistency of maximum log-likelihood means that we converge to the true latent causal variables up to permutation and scaling classes, in the limit of infinite data.

### A.5 IMPLEMENTATION DETAILS

#### A.5.1 EXPERIMENTS FOR FIGURE 5 (A)

**Data**  For experimental results of Figure 5 (a) the number of segments (*e.g.*,$\mathbf{u}$) $M$ is 10,20,30,40 and 50 respectively. For each segment, the number of latent causal variables is 2 and the sample size is 1000. We consider the following structural causal model

$$n_1 := \mathcal{N}(\beta(\mathbf{u})_{1,1}, \beta(\mathbf{u})_{1,2}), \qquad n_2 := \mathcal{N}(\beta(\mathbf{u})_{2,1}, \beta(\mathbf{u})_{2,2}), \qquad (49)$$

$$z_1 := n_1, \qquad\qquad\qquad z_2 := \lambda_{1,2}(\mathbf{u})z_1 + n_2, \qquad (50)$$

where we sample the mean $\beta(\mathbf{u})_{i,1}$ and variance $\beta(\mathbf{u})_{i,2}$ from uniform distributions $[-2, 2]$ and $[0.01, 3]$, respectively. We sample the weights $\lambda_{1,2}(\mathbf{u})$ from uniform distributions $[0.1, 2]$. We sample latent variable z according to Eqs. 49 and 50, and then mix them using a 2-layer multi-layer perceptron (MLP) to generate observed $\mathbf{x}$.

**Network and Optimization**  For all methods, we used a encoder, *e.g.* 3-layer fully connected network with 30 hidden nodes and Leaky-ReLU activation functions, and decoder, *e.g.* 3-layer fully connected network with 30 hidden nodes and Leaky-ReLU activation functions. We also use 3-layer fully connected network with 30 hidden nodes and Leaky-ReLU activation functions. For optimization, we use Adam optimizer with learning rate $1r = 1e - 3$. For hyper-parameters, we set $\beta = 4$ for the $\beta$-VAE. For CausalVAE, we set the hyper-parameters as suggested in the paper Yang et al. (2021).

#### A.5.2 EXPERIMENTS FOR FIGURE 5 (B) AND (C)

**Data**  For experimental results of Figure 5 (b) and (c), the number of segments is 30 and sample size is 1000, while the number (*e.g.*, dimension) of latent causal variables is 2,3,4,5 respectively. For

2-dimensional latent causal variables, the setting is the same as experiments for Figure 5 (a). For 3-dimensional case, we consider the following structural causal model:

$$n_1 := \mathcal{N}(\beta(\mathbf{u})_{1,1}, \beta(\mathbf{u})_{1,2}), \quad n_2 := \mathcal{N}(\beta(\mathbf{u})_{2,1}, \beta(\mathbf{u})_{2,2}), \quad n_3 := \mathcal{N}(\beta(\mathbf{u})_{3,1}, \beta(\mathbf{u})_{3,2}), \quad (51)$$
$$z_1 := n_1, \qquad\qquad z_2 := \lambda_{1,2}(\mathbf{u})z_1 + n_2, \qquad\qquad z_3 := \lambda_{2,3}(\mathbf{u})z_2 + n_3. \quad (52)$$

For 4-dimensional case, we consider the following structural causal model:

$$n_1 := \mathcal{N}(\beta(\mathbf{u})_{1,1}, \beta(\mathbf{u})_{1,2}), \qquad n_2 := \mathcal{N}(\beta(\mathbf{u})_{2,1}, \beta(\mathbf{u})_{2,2}), \qquad (53)$$
$$n_3 := \mathcal{N}(\beta(\mathbf{u})_{3,1}, \beta(\mathbf{u})_{3,2}), \qquad n_4 := \mathcal{N}(\beta(\mathbf{u})_{4,1}, \beta(\mathbf{u})_{4,2}), \qquad (54)$$
$$z_1 := n_1, \qquad\qquad z_2 := \lambda_{1,2}(\mathbf{u})z_1 + n_2, \qquad (55)$$
$$z_3 := \lambda_{2,3}(\mathbf{u})z_2 + n_3, \qquad z_4 := \lambda_{2,4}(\mathbf{u})z_2 + n_4. \qquad (56)$$

For 5-dimensional case, we consider the following structural causal model:

$$n_1 := \mathcal{N}(\beta(\mathbf{u})_{1,1}, \beta(\mathbf{u})_{1,2}), \quad n_2 := \mathcal{N}(\beta(\mathbf{u})_{2,1}, \beta(\mathbf{u})_{2,2}), \qquad (57)$$
$$n_3 := \mathcal{N}(\beta(\mathbf{u})_{3,1}, \beta(\mathbf{u})_{3,2}), \quad n_4 := \mathcal{N}(\beta(\mathbf{u})_{4,1}, \beta(\mathbf{u})_{4,2}), \quad n_5 := \mathcal{N}(\beta(\mathbf{u})_{5,1}, \beta(\mathbf{u})_{5,2}), \quad (58)$$
$$z_1 := n_1, \qquad\qquad z_2 := \lambda_{1,2}(\mathbf{u})z_1 + n_2, \qquad (59)$$
$$z_3 := \lambda_{2,3}(\mathbf{u})z_2 + n_3, \qquad z_4 := \lambda_{2,4}(\mathbf{u})z_2 + n_4, \qquad z_5 := \lambda_{4,5}(\mathbf{u})z_4 + n_5. \qquad (60)$$

Again, for all these Eqs 51-60, we sample the mean $\beta(\mathbf{u})_{i,1}$ and variance $\beta(\mathbf{u})_{i,2}$ from uniform distributions $[-2, 2]$ and $[0.01, 3]$, respectively. We sample the weights $\lambda_{i,j}(\mathbf{u})$ from uniform distributions $[0.1, 2]$. We sample latent variable z according to these equations, and then mix them using a 2-layer multi-layer perceptron (MLP) to generate observed $\mathbf{x}$.

**Network and Optimization** The network architecture, optimization and hyper-parameters are the same as used for experiments of Figure 5 (a), except for the 5-dimensional case where we use 40 hidden nodes for each linear layer.

## A.6 EXPERIMENTS ON CHANGES OF PART OF WEIGHTS

In this part, we conduct experiments on changes of part of weighs. To this end, we consider two cases on the following graph: $z_1 \to z_2 \to z_3$ (data details as mentioned in section A.5.2 for 3-dimensional case): *Change 1:* the weight on the edge of $z_1 \to z_2$ changes across $\mathbf{u}$, while the weight on the edge of $z_2 \to z_3$ remains unchanged across $\mathbf{u}$, *Change 2:* the weight on the edge of $z_1 \to z_2$ changes only for the first 15 segments (*e.g.*, $\mathbf{u}$), the weight on the edge of $z_2 \to z_3$ changes only for the last 15 segments (totally 30 segments). Again, the network architecture, optimization and hyper-parameters are the same as used for experiments of Figure 5 (a). The Figure 7 shows the MPC

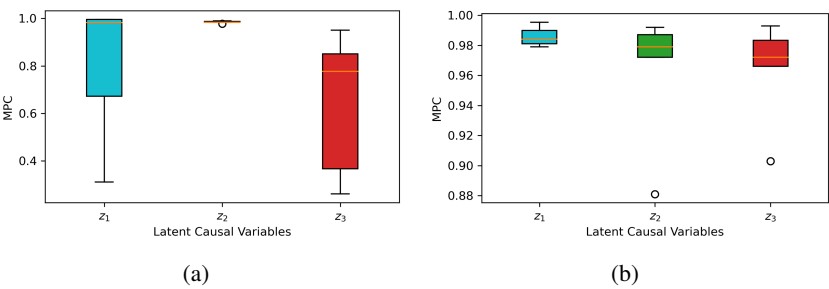

(a)           (b)

Figure 7: Performances of the proposed SuaVE in recovering the latent causal variables on *Change 1* (a), and *Change 2* (b).

between the recovered latent variables and the true ones by the proposed SuaVE. For *Change 1*, we can see that only the recovered $z_2$ obtains satisfactory MPC. For *Change 2*, all the recovered latent causal variables obtain high MPC, which seems to be promising to obtain identifiability. To further understand this interesting result, we investigate the recovered graph in the case of *Change 2*. Again, we also use a threshold to remove some edges to obtain the final graph. Since no identifiability is provided, we measure the SHD of CPDAG. We found that for *Change 2*, the proposed method obtains

**SHD = 1.0 ± 0.39**. Compared to that, we reported **SHD = 0.6 ± 0.35** in Figure 5 where the weights always change across $\mathbf{u}$. In summary, it seems to be promising to obtain identifiability up to Markov equivalence class for the setting of *Change 2*. It would be interesting to provide theoretical insights for this case in the future work.

### A.7 EXPERIMENTS ON I.I.D. $z_i$

As we mentioned in the main paper, our work is closely related to nonlinear ICA, and our assumptions includes the assumptions for identifying the Gaussian noise variables. To verify this point, we conduct experiments on the case where any two latent causal variables $z_i$ are independent given $\mathbf{u}$. Data details is the same as mentioned in section A.5.2, but we here enforce $\lambda_{i,j}(\mathbf{u}) = 0$ for each $i, j$. Again, the network architecture, optimization and hyper-parameters are the same as section A.5.2, except for learning rate (here we set $1r = 1e - 2$). Figure 8 shows the performances of the proposed SuaVE and iVAE. We can see that iVAE is slightly better than the proposed SuaVE, both obtain satisfying results in terms of MPC.

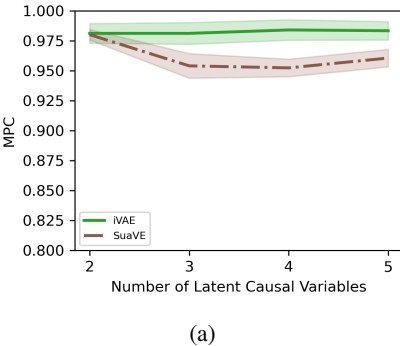

(a)

Figure 8: Performances of the proposed SuaVE and iVAE in recovering the independent latent causal variables conditional $\mathbf{u}$.

### A.8 DETAILED RESULTS ON FMRI

Figure 9 shows the recovered latent six signals (Blue) and the true ones (Red) within one day by the proposed SuaVE.

The recovered latent fMRI variables within one day by iVAE are depicted by Figure 10.

The recovered latent fMRI variables within one day by VAE are depicted by Figure 11.

The recovered latent fMRI variables within one day by $\beta$-VAE are depicted by Figure 12.

The recovered latent fMRI variables within one day by CausalVAE are depicted by Figure 13.

Implementation Details: For fMRI data, again, we used the same network architecture for encoder (*e.g.* 3-layer fully connected network with 30 hidden nodes for each layer) and decoder (*e.g.* 3-layer fully connected network with 30 hidden nodes for each layer) parts in all these models. For prior model in the proposed SuaVE and iVAE, we use 3-layer fully connected network with 30 hidden nodes for each layer. We assign an 3-layer fully connected network with 30 nodes to generate the weights to model the relations among latent causal variables in the proposed SuaVE. For hyper-parameters, we set $\beta = 4, 25, 50$ for $\beta$-VAE. For CausalVAE, we use the hyper-parameters setting as recommended by the authors (Yang et al., 2021), for generating observed $\mathbf{x}$, we use an invertible 2-layer multi-layer perceptron on the fMRI with different 5 random seeds.

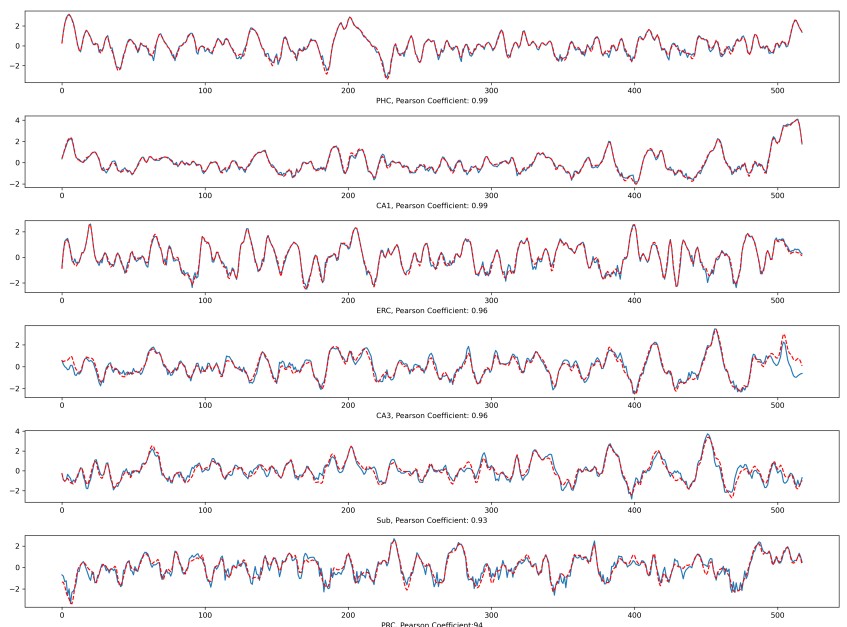

Figure 9: Recovered latent six signals (Blue) and the true ones (Red) within one day by the proposed SuaVE.

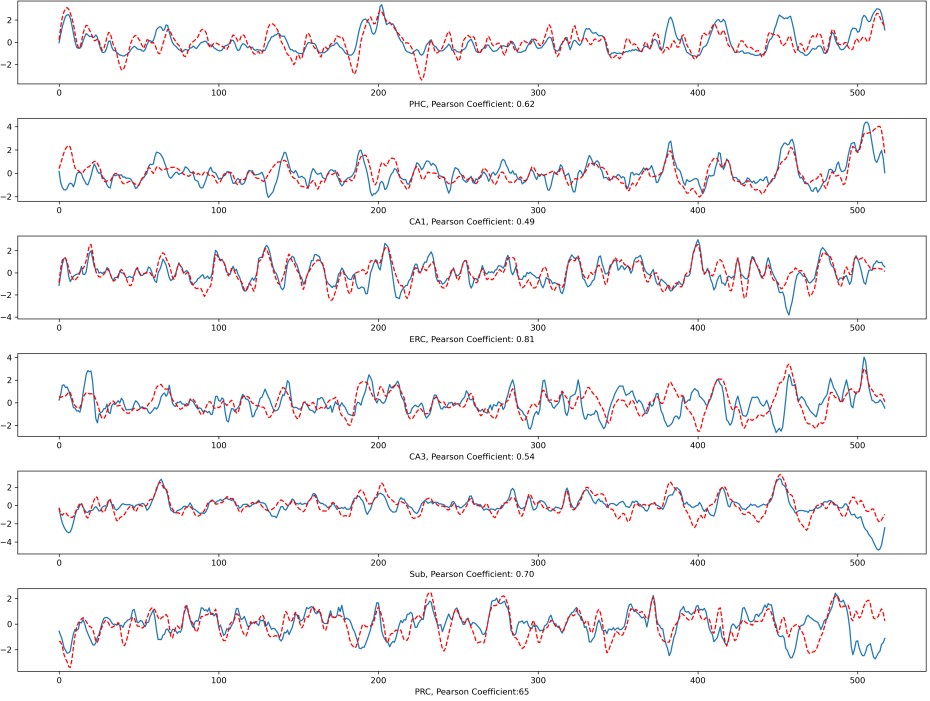

Figure 10: Recovered latent six signals (Blue) and the true ones (Red) within one day by iVAE.

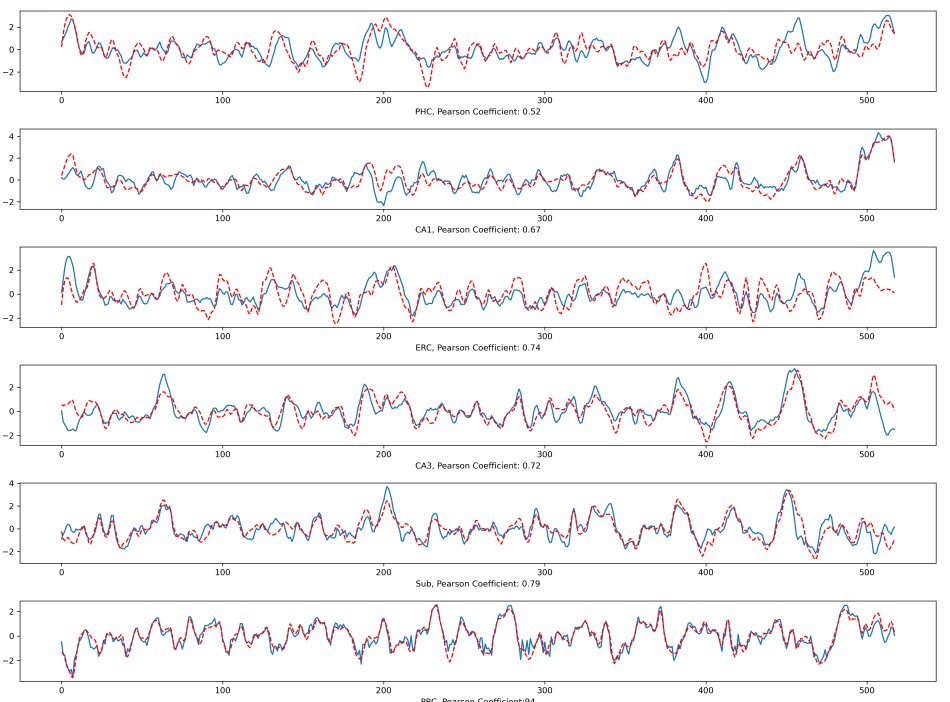

Figure 11: Recovered latent six signals (Blue) and the true ones (Red) within one day by VAE.

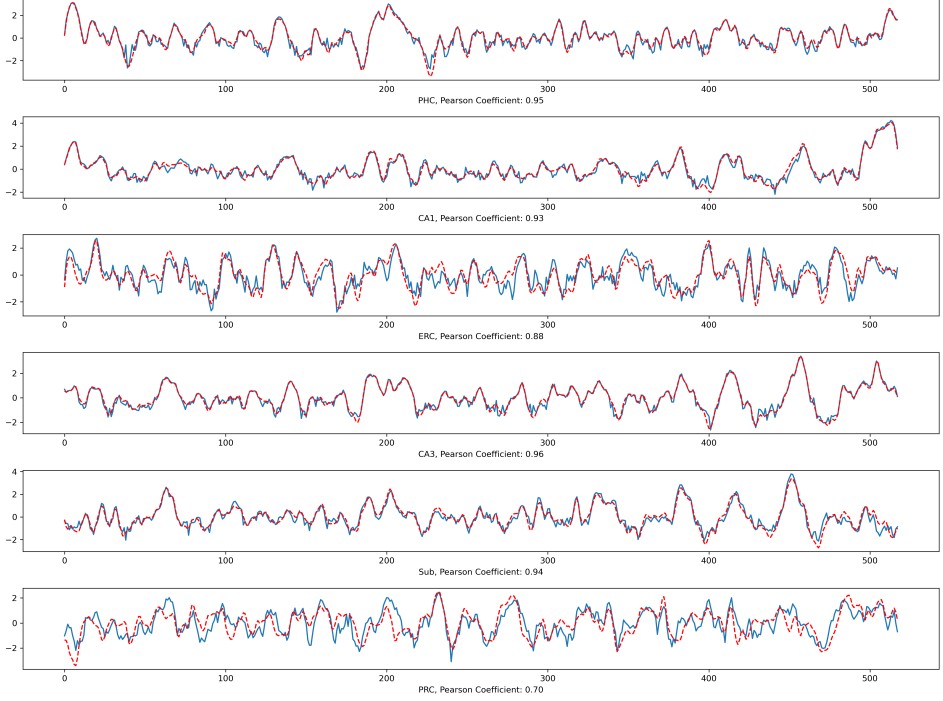

Figure 12: Recovered latent six signals (Blue) and the true ones (Red) within one day by $\beta$-VAE.

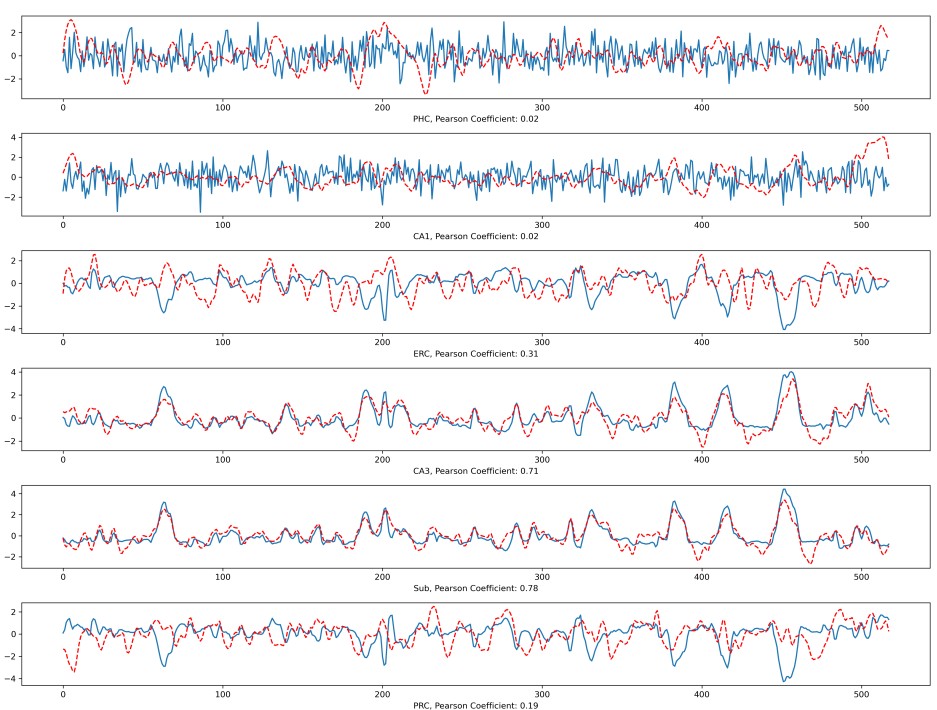

Figure 13: Recovered latent six signals (Blue) and the true ones (Red) within one day by CausalVAE.

