# OpenReview forum: "Identifying Weight-Variant Latent Causal Models"
_ICLR.cc/2023/Conference — Submitted to ICLR 2023_

### Official Review · Reviewer_FaeD · 2022-10-21

**Confidence:** 4
**Correctness:** 2
**Technical Novelty And Significance:** 3
**Empirical Novelty And Significance:** 3
**Recommendation:** 3

**Clarity, Quality, Novelty And Reproducibility:**

Novelty:
- To the best of my knowledge, the main theoretical result presented in this work is novel and not superseded by another theoretical work. I looked quickly at the proof and it seems sound (although some steps require clarification, see below). It reuses existing proof techniques but also introduces new ones.

Imprecise or wrong claims:
- In introduction: “A brief review is provided in Section 2. For the
supervised approach, where causal graphs or/and labels are known, the challenging identifiability
problem in latent space has been transferred to an identifiability problem in the observed space,
for which some commonly-used functional classes have been proven to be identifiable (Zhang &
Hyvarinen, 2012; Peters et al., 2014)”. I don’t understand how knowing the causal graph reduces the problem to causal discovery on observed variables… If the graph is known, there is no graph to learn, no?
- Section 3.1: “recent progress in nonlinear ICA (Hyvarinen et al., 2019; Khemakhem et al., 2020), which has shown that the independent latent noise variables n_i are identifiable under relatively mild assumptions, if they are modulated by an additionally observed variable u.” Please clarify that n_i are *conditionally* independent *given u*.
- Section 3.1: It is said that the n_i’s are identifiable, by recent nonlinear ICA advances. However, I think this is true only if the mapping from the n_i’s to the z_i (later referred to as g) is bijective. This should be made clear.
- Section 3 should be called “Three indeterminacies in latent causal models”, or something like this. The current title is confusing.
-Corollary 4.2: The introduction as well as the paragraph just before the corollary points toward the fact that the causal graph is identifiable, but the Corollary in itself guarantees only the identifiability of the MEC. The text should be adjusted to better reflect what is actually shown in the paper.

Clarity:
- Section 3.2: “According to the graph structure in the right column of Figure 1, assume
that z1 := n1, z2 := z1 + n2 and x := f(z1, z2) + ε (case 1).” But Figure 1 has only one column… Was it meant to point at figure 2? But the graphs of figure 2 do not correspond to what is described in words…
- Section 3.2: This section is really hard to follow. I believe the authors want to argue that two different causal models can yield the same distribution over observations (which is clearly true), but they provide virtually no explanation.
- Thm 4.1: Do we assume that k, the dimensionality of the minimal sufficient statistic is known?
- Thm 4.1: The statement “we can recover the latent causal variables z up to…” is too vague to be part of a theorem. What does it mean “to recover the latent causal variables”?
- Section 6.1: The authors refer to the Appendix for more details on how the data was generated, but the Appendix does not contain these details. I was trying to confirm that the ground-truth causal graph is truly complete right?
- The experimental section shows the SHD of the graph learned by SuaVE, but it is unclear how one goes from the parameter of SuaVE to an actual estimated graph. Is it by thresholding the learned parameter \lambda(u) to some value? How was this threshold selected? Did the authors consider adding regularisation to the learned graph? Without it, I suspect the graph would always be complete, no?

Proof of Thm 4.1:
- I am not sure I understand how the authors obtain equation (42). I quickly tried to show it myself using an argument similar to what is done for the other terms, but it appears to be more subtle. Given how crucial this step is to showing identifiability up to permutation, I believe this step should be made completely transparent.

Suggestions for experiments:
- It would be interesting to see what happens when the ground-truth graph is empty, which would correspond to a setting covered by iVAE’s theory, and thus it should perform well. Does SuaVE perform similarly to iVAE?
- CausalVAE performs very well (although not as good as SuaVE) on Figure 5a. The gap seems much more important for Figure 5b. Could the authors comment on that? How many latent causal variables are used in Figure5a?

Minor:
- In introduction: “...disentangled representation learning can be viewed as a special case of causal representation learning where the latent variables have no causal influences”. Many authors use the term disentanglement more broadly, without requiring the latent factors to be independent. Actually, Bengio et al (2013) made that point (see section 3.1 under “Simplicity of Factor Dependencies”).
- Related work: I disagree with the authors’ classification of Kuegelgen et al (2021) and Brehmer et al (2022) as “Supervised Causal Representation Learning”. The authors consider the knowledge of the graph as a supervision, which is true to some extent, but this kind of supervision is typically referred to as “weak”. Also, Brehmer et al (2022) does not assume the ground-truth causal graph is known.

**Strength And Weaknesses:**

Strengths:
- I follow rather closely this literature and I found the main theoretical contribution (Theorem 4.1) of this work to be novel and quite interesting. I believe other ICLR participants will be interested by this result.

Weaknesses:
- Many claims are either imprecise or wrong (see below).
- Some sections (especially Section 3.2) lack clarity (see below).
- Thm 4.1 should be contrasted with CausalVAE further, which shares many similarities with this work.
- Many experimental details are missing: For example, what is the number of latent causal variables used in Figure 5a? What is the number of segments available in Figure 5b? How’s the causal graph extracted from the SuaVE’s parameters to produce Figure 5c? This can be a problem for reproducibility.

**Summary Of The Paper:**

This paper proposes a solution to the problem of causal representation learning, which is about identifying latent variables and how they are causally related by observing only a nonlinear transformation of them. They propose a model in which the latent causal graph is linear gaussian and an auxiliary variable u is observed and has an influence on both the distribution of the gaussian noise and the coefficients of the linear models. They provide conditions under which this model is identifiable up to permutation and rescaling of the latent factors. Motivated by this result, they provide a method based on a simple variational lower bound to estimate the mixing functions. Compelling experimental results validates that the method can indeed identify the latent factors.

**Summary Of The Review:**

Although the identifiability result introduced in this work appears to be novel and very interesting, I cannot recommend acceptance due to (i) Imprecision/wrongness of certain claims, (ii) lack of clarity and (iii) the lack of details regarding some experiments. I strongly encourage the authors to improve the writing of this work, since I believe the main theoretical result would be of interest to the community.

---

> ### Author Response · Authors · 2022-11-16
> **Rebuttal**
>
> __*Q1: If the graph is known, there is no graph to learn, no?*__
> We sincerely thank the reviewer for this suggestion. We have modified this point, as shown in the red words in the new rebuttal version.
>
> __*Q2: Please clarify that ni are conditionally independent given u.*__
> We sincerely thank the reviewer for this suggestion. We have clarified this point in the rebuttal version.
>
> __*Q3: I think this is true only if the mapping from the ni’s to the zi (later referred to as g) is bijective. This should be made clear.*__
> We sincerely thank the reviewer for this suggestion. Please see in the red words in section 3.1 in the new rebuttal version. Due to DAG constraint in causal models and linear models from n to z, the condition that the mapping is bijective can be ensured in the proposed weights-variant models.
>
> __*Q4: Section 3 should be called “Three indeterminacies in latent causal models”*__
> We sincerely thank the reviewer for this suggestion. And all property or properties in original version are replaced by indeterminacy or indeterminacies.
>
> __*Q5: the Corollary in itself guarantees only the identifiability of the MEC*__
> Thank you for your careful reading. We have updated the text to emphasize that it is the MEC, instead of the entire DAG, that is identifiable.
>
> __*Q6: Section 3.2: Figure 1 has only one column, the section is really hard to follow*__
> We sincerely thank the reviewer for pointing this out. “According to the graph structure in the right column of Figure 1" should be “According to the graph structure in Figure 1", which makes the section hard to follow. We have modified this point.
>
> __*Q7: Thm 4.1: Do we assume that k, the dimensionality of the minimal sufficient statistic is known?*__
> In our Thm 4.1, we assume that the number of $n_i$ (or $z_i$) is known. Since we argue that we do not give assumption on graph structure, we do not assume that k is known. But if we give the number of $n_i$ (or $z_i$), k has been limited in $2n \leq k \leq n + (n*(n+1))/2$, as we mentioned in the manuscript.
>
> __*Q8: Thm 4.1: The statement “we can recover…” is too vague to be part of a theorem*__
> We sincerely thank the reviewer for pointing this out. We have modified this point, 'then the parameters $\mathbf{f}, \lambda, \beta$ are identifiable up to the equivalence class with permutation and linear scaling.
>
> __*Q9: More details on how the data was generated*__
> We sincerely thank the reviewer for pointing this out. We have added a new section to clarify these details, please see A.5 in Appendix for details.
>
> __*Q10: Is it by thresholding the learned parameter $\lambda(u)$ to some value? How was this threshold selected? Did the authors consider adding regularisation to the learned graph? Without it, I suspect the graph would always be complete, no?*__
> We use a threshold on the learned parameter $\lambda(u)$. We empirically select the threshold according to the groundtruth. Usually, we set threshold = 0.1, but sometimes threshold = 0.01 due to the scaling, Indeed, due to finite data and challenge in optimization, we obtain complete graphs before thresholding the learned parameter $\lambda(u)$. We have considered sparsity based regularisations, e.g., L1 or Bayesian compression for deep learning, but these regularisations may also introduce new problems, i.e., 1) how to decide the hyper-parameter for the regularisation term to balance reconstruction and sparsity, 2) Even with sparsity based regularisations, we may still need a threshold to exactly obtain zero-value weights, since sparsity based regularisations often results in weights with low values, not exact zero values. We mainly focus on identifiability in this work, designing more effective method than the proposed method is interesting and an open problem, under our model assumptions.
>
> __*Q11: How the authors obtain equation (42).*__
> We really thanks the reviewer for checking this step. We carefully checked equation (42) in the original version, and found that this is a nontrivial step, due to a special case: $\lambda_{i,j}(\mathbf{u})= {\hat \lambda_{i,j}}({\mathbf{u}}) +b$ where $b$ is a non-zero constant. For example, if $z_2=(\hat \lambda_{1,2}(\mathbf{u})+b)z_1+n_2$, then the term $bz_1$ is still unchanged across all $\mathbf{u}$, and thus can be 'absorbed' by the nonlinear mapping from latent variables $\mathbf{z}$ to the observed variable $\mathbf{x}$ due to the transitivity. Therefore, we have a new assumption: assumption (iv) as shown in Thm.1 in the new rebuttal version, to handle the special case (see further discussions and assumption (iv) in section A.2). And we have added a new proof step for permutation, please see step IV in the new rebuttal version.
>
> __*Q12:  Results when the ground-truth graph is empty*__
> We have added a new section A.7 to report this special case. In addition, we also added a new section A.6 to show experiments on
>  the changes of part of weights. Please see these two sections for further discussions.

---

> > ### Comment · Reviewer_FaeD · 2022-11-23
> > **Response**
> >
> > I thank the authors for considering my points seriously.
> >
> > **Proof of Theorem 4.1:**
> >
> > I just noticed a problem in equation (35) and (36) (new manuscript). The matrix product $A^{(3)}\hat{z}_{ij}$ is not spelled out correctly. My understanding is that the shape of $A^{(3)}$ and $A^{(6)}$ should be $n(n-1)/2 \times n(n-1)/2$, but it seems the authors assume it is $n\times n$. In fact, both (35) and (36) should contain terms with $\hat{z}_k\hat{z}_j$ where $k \not= i \not= j$. Unfortunately I don't have enough time to work out the consequences of this mistake and whether it can be fixed. I will read the additional steps of the proof once the above point is clarified.
> >
> > **Extra assumption**
> >
> > I'd like to comment on the newly introduced assumption of Thm 4.1. It appears to be very strong in my opinion, it basically states that there exists a "u" such that all latent variables are mutually independent. Not only that, but that the learner *knows* which "u" corresponds to this extreme case. Is this a correct interpretation? If so, I believe it makes the result a bit less appealing.

---

> > > ### Author Response · Authors · 2022-11-24
> > > **Response**
> > >
> > > We sincerely thank the reviewer for these comments.
> > >
> > > __*Proof of Theorem 4.1:*__
> > > The shape of  $A^{(3)}$ and $A^{(6)}$ should be $n \times n(n-1)/2$. In the final version we will fix both by replacing both by ${\mathbf{A}}_{i,:}^{(3)} \mathbf{\hat z}\_{i,j}$ and ${\mathbf{A}}\_{i,:}^{(6)} \mathbf{\hat z}\_{i,j}$, where $\mathbf{\hat z}\_{i,j}$ is defined in Eq. (30). These typoes will not affect the following proof.  In addition, we note that there is a typo in Eq. (45), it should be $(\mathbf{B}\mathbf{P}-\mathbf{ A}^{(1)} \mathbf{\hat B})\mathbf{\hat n} = \mathbf{ c}^{(1)}-\mathbf{B}\mathbf{ c}_n$. Thank you for your careful reading.
> > >
> > > __*Extra assumption*__
> > > The extra assumption (iv) is to avoid a special case: $\lambda_{i,j}({\mathbf{u}}) = \lambda'_{i,j}{\mathbf{u}} +b $ where $b$ is a non-zero constant. Please see the paragraph 'Changes of weight and Assumption (iv)' in A.2 for details. To avoid the case, we give the extra assumption (iv). In fact, we can further relax it by replacing it by a new assumption that $\exists \mathbf{u}'$, $\lambda\_{i,j}({\mathbf{u}'})=0$ (i.e., all latent variables are mutually independent). We will replace the extra assumption (iv) by the new assumption in the final version. The new assumption is only to restrict the function class for $\lambda\_{i,j}$. That is, it do not require that the observed data include the data corresponding to the $\mathbf{u}'$. And we do not require to know which $\mathbf{u}'$. The new assumption is also to handle the special case, so that the function class for $\lambda\_{i,j}$ can not be separated as that a new $\hat \lambda\_{i,j}$ plus a non-zero constant. It is open to further relax the new assumption to handle the special case.

---

> > > > ### Comment · Reviewer_FaeD · 2022-12-02
> > > > **Response**
> > > >
> > > > It looks like indeed the mistake (I wouldn't call it a typo) doesn't affect the rest of the argument.
> > > >
> > > > I thank the authors for their clarifications regarding the extra assumption. I still believe the assumption is rather much stronger.
> > > >
> > > > However, in the proof of thm 4.1, the application of Sorrenson's result is unclear. The updated proof mentions the assumptions of Sorrenson's are satisfied in Thm 4.1, but it's not clear which result in Sorrenson et al. (2020) the authors are referring to. My guess is that they make a correspondence between n (in their work) and z (in Sorrenson's). But the correspondence just doesn't hold since (i) the function that maps the noises n to the latent z might not be invertible and (ii) this same map depends on u, and thus the map from the noise n to x also depends on u, which violates the assumption in Sorrenson et al. (2020). Of course this is all based on a guess about what the authors meant, but what is certain is that the argument is not transparent.
> > > >
> > > > The main strength of this work was its theoretical result. But, unfortunately, the more I engage with its proof, the more I doubt its validity. This forces me to reduce my score to 3, since this work is not ready for publication yet. I encourage the authors to clarify the argument in the proof before resubmitting.

---

> > > > > ### Author Response · Authors · 2022-12-03
> > > > > **Thank you again for your further comments, and A kind request for further feedback for proof of thm 4.1**
> > > > >
> > > > > __*the extra assumption*__: We would like to clarify our understanding about the extra assumption from a high-level viewpoint: suppose that the causal influences (or weights) include two part: part (I) is depending on u (e.g., $\lambda'(\mathbf{u})$ ), while part (II) is not (e.g., b). In this case, part (I) will not be 'absorbed' by the mapping from z to x, while part (II) is invariant across u, and thus will be 'absorbed'. As a result, part (II) is not identifiable. Thus, we believe that there is a restriction for the function class of $\lambda$. Loosely speaking, the function class should not contain a constant term. To formulate the function class more rigorously, suppose that $\lambda$ can be approximated by a Taylor series. Therefore, we enforce the extra assumption, e.g.,  $\exists \mathbf{u}'\_{i,j}$, $\lambda\_{i,j}({\mathbf{u}'\_{i,j}})=0$. We would also like to emphasise that:
> > > > >
> > > > > It do not require that there exists a __common__ $\mathbf{u}'$ for which all latent variables are mutually independent. For different $\lambda\_{i,j}$ , $\mathbf{u}'\_{i,j}$ could be different. For example, for $\lambda\_{1,2}$, $\mathbf{u}\_{1,2}'$ could be [0,0,1] so that $\lambda\_{1,2}([0,0,1])=0$, while for $\lambda\_{1,3}$, $\mathbf{u}'\_{1,3}$ could be [0,1,0] so that $\lambda\_{1,3}([0,1,0])=0$. (__This is a very important point, which is neglected in our previous response. We hope this response properly addresses your concern about the assumption.__)
> > > > >
> > > > >
> > > > > __*Sorrenson's result:*__  We would like to emphasize the following point first. This work is to leverage nonlinear ICA to identify z. Clearly, the assumptions of thm 4.1 reduces to the assumptions of identifying n in Gaussian case, if there are no edges among z. That is, the assumptions of thm 4.1 are stronger than the assumptions of identifying n in Gaussian case. This is further verified by the experimental results in section A.7, where we can see that the proposed method obtains satisfactory perform when there are no edges among z. Therefore, we argue that our assumptions in Theorem 4.1 include the assumptions for identifying the Gaussian noise variables.
> > > > >
> > > > > For __(i) the function that maps the noises n to the latent z might not be invertible__ As we mentioned in __*Q3*__ in the initial response__, due to DAG constraint in causal models and linear models from n to z, the mapping for n to z must be invertible across u.
> > > > >
> > > > > For __(ii) the map from the noise n to x also depends on u, which violates the assumption in Sorrenson et al. (2020)__
> > > > >
> > > > > To clarify this, let us consider the effects of the mapping from n to x for the proof of Sorrenson et al. (2020).
> > > > >
> > > > > The effects of the mapping for the proof in Sorrenson et al. (2020) include two parts: __*part (a)*__ the mapping is invertible, so that the PDF of x conditional on u can be given by the PDF of n conditional on u and the log of Jacobian determinant, as mentioned in Eq. (10) in Sorrenson et al. (2020). __*part (b)*__ the corresponding log of Jacobian determinant do not depend on u (since the mapping from n to x do not depend on u in Sorrenson et al. (2020)), thus it can be removed by different u, as mentioned in Eq. (11) in Sorrenson et al. (2020). Except for these two steps, the remaining proof of in Sorrenson et al. (2020) do not depend on the mapping.
> > > > >
> > > > > Now, let us consider the mapping from n to x in our case to show that it do not affect the above two steps.
> > > > >
> > > > > Denote by $logJ$ the log of Jacobian determinant, which is related to the mapping from n to x.
> > > > >
> > > > > Denote by $J_n$ the Jacobian determinant, which is related to the mapping from n to z.
> > > > >
> > > > > Denote by $J_z$ the Jacobian determinant, which is related to the mapping from z to x.
> > > > >
> > > > > Since the mapping from n to x is the composition of the invertible mapping from n to z (as mentioned above) and the invertible mapping from z to x (According to assumption (ii)), we have: the mapping n to x is invertible, __*arriving at the above part (a)*__. In addition, for the composition, we have $log|J| =log|J_zJ_n|=log(|J_z||J_n|)$. Clearly, for $|J_z|$, it do not depend on u (since the mapping from z to x do not depend on u.). Although the mapping from n to z depends on u, $|J_n|$ do not depend on u. The reason is due to DAG constraint in causal models and the assumption of linear model among z. For example, $ \mathbf{z =B(u)n}$, since DAG constraint, the matrix $\mathbf{B(u)}$ is a triangular matrix for the correct order and the main diagonal are 1, we have $|J_n|=1$, which is not affected by permutation of $\mathbf{B(u)}$. As a result, $log|J| =log|J_z|$ do not depend on u (__*Arriving at the above part (b)*__ ), thus can be removed by different u, similar to the step of Eq. (11) in Sorrenson et al. (2020). Therefore, the mapping from n to z that depends on u, do not affect the proof in Sorrenson et al. (2020).
> > > > >
> > > > > A kind request for further feedback for your concern about the proof. We understand you are very busy and appreciate your time. Your feedback is valuable to us.

---

> ### Author Response · Authors · 2022-11-16
> **Rebuttal**
>
> __*Q13: CausalVAE performs very well (although not as good as SuaVE) on Figure 5a.*__
> Figure 5 a is based on 2-dimensional $\mathbf{z}$. We have clarified the experimental details in section A.5.1. Note that CausalVAE have the additional operator: the mapping from n to z. In this special case, due to allowing dependence among latent variables, CausalVAE may be better than ivae, betavae and vae. But we can see that CausalVAE have no such advantage when the number of latent variable is large, see Figure 5b, due to non-identifiability.
>
> Once again, we appreciate your time devoted to reviewing this paper.

---

### Official Review · Reviewer_P5md · 2022-10-24

**Confidence:** 2
**Correctness:** 3
**Technical Novelty And Significance:** 2
**Empirical Novelty And Significance:** 2
**Recommendation:** 5

**Clarity, Quality, Novelty And Reproducibility:**

The paper is clear and I particularly enjoyed reading the background on identifiability issues in latent models. I don't have the expertise to truly understand how novel the identifiability result is in relation to existing literature, but on a causal and practical side my view is that this method is unlikely to be useful.

**Strength And Weaknesses:**

**Strengths**
- The paper is clearly written. The authors give a lot of details into why latent structure is not identifiable which serves to motivate the proposed approach. Their model is general enough that it can be a plausible explanation for latent structure in practice.

**Questions / comments**
- It is not clear what the authors are identifying in Corollary 4.2. When marginalizing over latent variables, the structure is a mixed graph and not a DAG, and thus the equivalence class is a PAG. Moreover with the assumed model in Sec. 4, all variables Z are dependent so the distribution of the data really doesn't narrow the class of compatible models.
- The graph in Fig. 4 is non-standard, if z_i is a function of u, then u should have an arrow into z_i. Is there any difference between Eq. (2) and $z_i := g(\mathbf u,\mathbf z) + n_i$?
- Thm. 4.1 depends on a lot of unverifiable, and one may argue, unrealistic assumptions about the latent structure. I understand that these may be necessary from a theoretical standpoint but also makes this method inapplicable in practice. At the very least one needs sensitivity analysis to understand whether this method can be used in practice. The current experiments are probably insufficient to understand the proposed approach.

**Summary Of The Paper:**

Observed signals are often generated from latent variables with significant structure that can provide significant understanding into the dynamics of the phenomenon under investigation. This structure is often unidentifiable without strong prior assumptions. The authors study a particular model, in which noise is Gaussian and associations between latent variables are linear, and show that some aspects of the latent structure are identifiable up to some transformations.

**Summary Of The Review:**

Well written paper with potentially some interesting results on latent model identifiability. The assumed structure however is quite restrictive and it is not clear what the effect of these assumptions are for performance in practice. Since this method is presented as a contribution to the causal literature I don't think it is likely to have a strong impact, simply because it relies on too many unverifiable assumptions not only on the causal structure of the underlying system but also on the functional dependencies and distributions of all variables.

---

> ### Author Response · Authors · 2022-11-16
> **Rebuttal**
>
> __*Q1: It is not clear what the authors are identifying in Corollary 4.2*__
> Note that here $\mathbf{u}$ is observed, so we did not marginalize over latent variables. Theorem 4.1 has shown that the latent causal variables $\mathbf{z}$ are identifiable, so identifiability of causal structure in latent space reduces to the identifiability of the causal structure in observed space. Moreover, although $\mathbf{u}$ is the confounder of $\mathbf{z}$, $\mathbf{u}$ is an observed variable. Therefore, we can recover the Markov equivalence class over $z$.
>
> __*Q2: The graph in Fig. 4 is non-standard, if $z_i$ is a function of u, then u should have an arrow into $z_i$. Is there any difference between Eq. (2) and $z_i=g(u,z)+n_i$?*__
> 1) We use 'red' line to distinguish the changes of weights across $\mathbf{u}$ from the traditional definition of edges (e.g., black lines in Figure 4) in standard SCM. If we use edges from $\mathbf{u}$ to $z_i$, this may also lead to confusion, since this means that $\mathbf{u}$ is a parent node of $z_i$ in standard SCM. Based on this point, we use 'red' line from $\mathbf{u}$ to edges to indicate that the weights among latent causal variables is changed by $\mathbf{u}$, as mentioned in Eq. (2).
> 2)  Eq. (2) is different with $z_i=g(u,z)+n_i$ in that: 1) Eq. (2) is a linear model, 2)Even if g here denotes a linear model, in traditional SCM, $z_i=g(u,z)+n_i$ could be regarded as $z_i=W[u,z]$, where u is the parent node. But in this work, $\mathbf{u}$ just affects the edges from the parent nodes to $z_i$. Therefore, to clarify this point, we use 'red' lines.
>
> __*Q3: Thm. 4.1 depends on unrealistic assumptions. Experiments are probably insufficient*__
> For model assumptions, latent causal discovery is a notoriously hard problem that requires fundamental theoretical advances before practical algorithms can be developed. Identifying latent causal graph is generally not possible without certain assumptions. This work is a new research line to explore the relation between change of causal influences and identifiability of latent causal discovery. Experiments on simulation data and fMRI data verify the theoretical results. We believe that this might be a significant source of inspiration, and promote causal representation learning in real applications. We also report new experimental results to give further discussions for the changes of part of weights and i.i.d., $z_i$, please see appendix A.6 and A.7 for more details.

---

### Official Review · Reviewer_2uCg · 2022-10-24

**Confidence:** 3
**Correctness:** 3
**Technical Novelty And Significance:** 3
**Empirical Novelty And Significance:** 3
**Recommendation:** 8

**Clarity, Quality, Novelty And Reproducibility:**

Overall, I think this paper is very clearly written, and the quality is good. Also, it contains novel theories on the problem. Some questions that helps improving the paper are the following:

**Questions.**
1. In what practical scenarios are we given an observed variable U?
2. I don’t understand what the red edges are in Figure 4. In the standard structural causal models, there are no edges from variables to edges.
3. Is the assumption 2 in Theorem 4.1. can be held in practice? If X=x is a binary or discrete variable and the function f is bijective, then the latent variable Z must also be binary or discrete. Therefore, in which scenarios can we think that assumption 2 holds?

**Details Of Ethics Concerns:**

This paper doesn't contain ethical concerns.

**Strength And Weaknesses:**

## Strength
1. This paper is very well and claerly written.
2. A detailed and interesting motivation in Section 3 helps a lot in understanding the problem.
3. This paper contains extensive simulation results.


## Weakness
1. More background information for understanding the paper (e.g., what’s nonlinear ICA?) could help.
2. More detailed example in the problem of causal representation learning (e.g., why do we want to learn the relations between variables Zi?) is needed.

**Summary Of The Paper:**

This paper provides a theory for identifying latent causal structure using observational data when the weight-variant linear gaussian model assumption holds.

**Summary Of The Review:**

Overall, I think this paper is very clearly written, and the quality is good. Also, it contains novel theories on the problem. I think the paper can be improved by if more preliminaries and detailed example of causal representation learning are provided.

---

> ### Author Response · Authors · 2022-11-16
> **Rebuttal**
>
> __*Q1: In what practical scenarios are we given an observed variable U?*__
> This auxiliary variable could represent time indices, domain indices, or almost any additional or side information making it applicable to diverse types of problems and applications. More specifically, we expect that this work can be extended to domain adaptation, for which $\mathbf{u}$ could be domain indices to denote different domains.
>
> __*Q2: I don’t understand what the red edges are in Figure 4*__
> Good question, in standard SCM there is no edge from nodes to edges. We also consider this point in our manuscript. Therefore we use 'red' line to distinguish the changes of weights across $\mathbf{u}$ from the definition of edges (e.g., black line in Figure 4) in standard SCM. If we use edges from $\mathbf{u}$ to $z_i$, this may also lead to confusion, since this means that $\mathbf{u}$ is the parent node of $z_i$ in standard SCM. Based on this point, we use 'red' line from $\mathbf{u}$ to edges to indicate that the weights among latent causal variables are changed by $\mathbf{u}$, as mentioned in Eq. (2).
>
> __*Q3: Is the assumption 2 in Theorem 4.1. can be held in practice?discrete case?*__
> This work only consider continues domain, not discrete domain. In this case, bijective could be easily satisfied, \eg flow based models. For latent variables models, it is common to assume the mapping to be bijective to obtain identifiability.

---

### Official Review · Reviewer_DXAx · 2022-10-24

**Confidence:** 5
**Correctness:** 2
**Technical Novelty And Significance:** 2
**Empirical Novelty And Significance:** 2
**Recommendation:** 3

**Clarity, Quality, Novelty And Reproducibility:**

The exposition is very clear. Novelty is not too high given the existing work in nonlinear ICA. This is not a problem though.

**Strength And Weaknesses:**

There are certain inconsistencies in the assumptions that need to be addressed by the authors.

The assumptions might be a bit too strong/unrealistic. This is evidenced by the fact that the equivalence class reduces to a single graph given the specified assumptions. If the authors showed that these assumptions are also necessary, I think that would be an interesting contribution. But I don't see such an argument.

**Summary Of The Paper:**

The authors extend the nonlinear ICA results on identifiable latent factors to when the latent factors have a causal structure.

**Summary Of The Review:**

Thank you for your submission. I have some major concerns below. I would be happy to update my score based on the authors' clarifications/ addressing of these issues.

1. The discussion in the first paragraph of Section 3.1 is inconsistent with the causal graph given in Figure 1: It is extremely unlikely that n_i's are independent if they are all caused by the same variable u. This assumption that n_i's are independent seems to be necessary to make the nonlinear ICA framework applicable here so this issue should be addressed by the authors.

2. The result inherits the proof of Khemakhem et al. Therefore it inherits some of the the strict assumptions of their theorem. For example, the proof, when one looks closely, assumes that for any two causal models, the noise distributions have to be identical. This is quite a strong assumption and needs to be mentioned explicitly. I recommend the authors explicitly list the set of assumptions including this one in their manuscript.

3. The assumption that U is a variable that is observed, and it can be used to "intervene" on every edge, i.e., every configuration of latent variables and also several times as required by the theorem might be a strong assumption. It might be better to dissect this assumption a bit more to justify when it can be achieved in practice.


Minor Comments and Suggestions:

The main contribution is in Section 4. Although Section 3 gives a nice exposition with examples, the observations here are not surprising that the mapping can absorb such indeterminacies.

"causal supergraph"
I don't believe there is a need for a new name here. This fully connected DAG is known as a tournament in graph theory. Also known as a total order.

"To address this issue, we allow causal influences among latent
causal variables to be modulated by an additionally observed variable"
the word "modulated" is not well-defined here.

---

> ### Author Response · Authors · 2022-11-16
> **Rebuttal**
>
> __* Q1: $n_i$ are independent if they are all caused by the same variable u*__
> As we mentioned in the original, we assume that the variables $\mathbf{u}$ is observed, thus'$n_i$ are mutually independent' implicitly means that '$n_i$ are mutually independent conditional on the observed $\mathbf{u}$', similar to nonlinear ICA. We have clarified this point in the rebuttal version. We sincerely thank the reviewer for this suggestion.
>
> __*Q2: the noise distributions have to be identical, list the set of assumptions*__
> First, we clarify that our understanding about the 'identical' here is considering that any two possible solutions have to be the same distribution family, not the parameters of distribution family. Currently, it is common to give assumption on the distributions of latent variables to limit solution space and thus obtain identifiability. Eq. (1) has clearly shown that we assume any two possible solutions to be the same distribution family. And if we assume the distributions, the proof process also needs to assume the distributions to be identical to meet the assumption. Therefore, we think that in the proof process, the identical noise distributions is reasonable. For further understanding and clarifying the assumptions, we introduce a new section in APPENDIX, please see section A.2 for further discussions.
>
> __*Q3: The assumption that U is a variable that is observed, and it can be used to "intervene" on every edge*__
> For the model assumptions, latent causal discovery is a notoriously hard problem that requires fundamental theoretical advances before practical algorithms can be developed. Identifying latent causal graph is generally not possible without certain assumptions. This work is a new research line to explore the relation between change of causal influences and identifiability of latent causal discovery. Experiments on simulation data and fMRI data verify the theoretical results. We believe that this might be a significant source of inspiration, and promote causal representation learning in real applications. In addition, we report new experimental results to give further discussions for the changes of part of weights, please see A.6 for details.

---

> > ### Author Response · Authors · 2022-12-04
> > **Could you please provide your further feedback on our response?**
> >
> > Dear Reviewer DXAx,
> >
> > Thanks for providing the initial comments. We have provided responses to your main concerns. If there is any other concern, please let us know, and we will immediately respond to that. We understand you are very busy and appreciate your time. Your feedback is valuable to us. Thank you.

---

### Official Review · Reviewer_kexo · 2022-10-25

**Confidence:** 4
**Correctness:** 3
**Technical Novelty And Significance:** 3
**Empirical Novelty And Significance:** 3
**Recommendation:** 6

**Clarity, Quality, Novelty And Reproducibility:**

Clarity and quality: the paper would benefit more from adding motivations for the model structures, not just for the sake of identifiability but also its usefulness in practical scenarios.

Novelty: see above.

**Strength And Weaknesses:**

Utilizing recent advances in nonlinear ICA identifiability to propose new causal representation learning algorithms is an interesting point of view. This paper is very well-written and I expect the results can make novel contribution to the field. I also like the way that the authors explain the non-identifiability of latent causal representations, motivates how to tackle them, and avoid the constraint-based optimization of no-tears by pre-defined causal orders.

Compared to other reviews, I might be less worried about the empirical evaluations.  Based on my own understanding the technical details, all the proofs seems to hold except Corollary 4.2.

- The main issue is that I am not convinced how Corollary 4.2 can justify the identification algorithm proposed by the paper. To be more specific, Corollary 4.2 claims that the linear scaling of latent variables does not affect the identifiability of the latent causal structure. This only true if the identification is based on e.g. conditional independence relationships. However, this is not what the learning algorithm proposes. As I understand, SuaVE implicitly models the adjacency matrix via continuous causal weights + thresholding pruning, which I guess is inspired by the parameterization used in No-tears and many other causal discovery papers. This is fine when the dataset is fully observed. However, in the case of SuaVE, $z$ can be only recovered up to linear scaling. Therefore, the scaling of $\lambda(u)$ *will* get affected by linear scaling of $z$. This is especially the case since the authors also uses a non-zero thresholding (0.1) to round up the weights; which essentially means that this threshold also needs to be scaled proportionally to the linear scaling of $z$, which is very tricky and not considered by the paper. Also, the $u$-dependent parameterization for causal weights $\lambda(u)$ introduces additional issues since the existence of the corresponding edges might depend on the values of $u$ as well, which is not dealt with in Huang et al. 2020. I guess the above issue might be avoided by an alternative parameterization, in which you could decouple the parameterization of causal graphs and the edge functions (as in SCMs), where the causal graph can be modeled as via binary variables, that can be used to 0-1 mask the functions of its parent nodes.

- The other concern is I am not sure how to justify the modelling choice of SuaVE (Figure 4) and how reasonable this assumption is in the real-world causal representation learning applications. To me it seems a bit unintuitive to treat the latent variables in i-VAEs as non-exogenous noises here ($n$); also the fact that the latent causal graph might vary across data points seems like a non-necessary complication. Could the authors provide some examples to justify those assumptions?

**Summary Of The Paper:**

This paper proposed SuaVE, a new identifiable model for causal representations learning. SuaVE is an extension of i-VAE model in the sense that: 1, an additional latent causal variables layer is generated conditioned on the latent noise variables;  2, the latent causal variables are modelled via linear Gaussian models with $u$-dependent causal weights. Under certain conditions, the authors shows that 1), the latent causal variables can be identified up to permutation and linear scaling; 2), the causal graph among latent causal variables can be identified up to its Markov equivalence class. The authors then proposed practical algorithms for identifying the latent causal variables and conducted both synthetic and real-world experiments.

**Summary Of The Review:**

This is an interesting paper that addresses an important question in causal representation learning; and seems to possess enough novelty to the best of my knowledge. However, due to my certain technical concerns, I can only recommend a weak accept.

---

> ### Author Response · Authors · 2022-11-16
> **Rebuttal**
>
> __*Q1:How Corollary 4.2 can justify the identification algorithm proposed by the paper*__
> First, since we are concerned with linear Gaussian models over the latent variables, the identifiability up to equivalence class also holds for score-based methods that use the likelihood of data as objective functions. This is because independence equivalence, i.e., two DAGs have the same conditional independence relations, is the same as distributional equivalence for linear-Gaussian models, i.e., two Bayesian networks corresponding to the two DAGs can define the same probability distribution. That is to say, score-based methods also find the structure based on independence relations implicitly. Because the scaling does not change the independence relations, it will also not affect the identifiability of the graph structure. Also, note that Corollary 4.2 is about theoretical identifiability, and it does not rely on the estimation algorithm and the chosen threshold. On the other hand, in practical estimation, for fMRI data, we found using the same threshold (e.g.,0.1) to remove edges is reasonable; for experiments in Figure 5(b), we set threshold=0.1 as well in most cases. We agree that different scaling and thresholds may influence the estimated causal structure. To mitigate this issue, we normalize the causal strength. We mainly focus on identifiability in this work, and designing a more effective method is interesting and is our next step.
>
> __*Q2: How to justify the modelling choice of SuaVE*___
> For the model assumptions, latent causal discovery is a notoriously hard problem that requires fundamental theoretical advances before practical algorithms can be developed. Identifying latent causal graph is generally not possible without certain assumptions. This work is a new research line to explore the relation between change of causal influences and identifiability of latent causal discovery. Experiments on simulation data and fMRI data verify the theoretical results. We believe that this might be a significant source of inspiration, and promote causal representation learning in real applications. For further understanding the assumptions, we introduce a new section in APPENDIX, please see section A.2 for further discussions. For the necessity of latent causal graph might vary across data points: to align the following discussion, we first clarify that we assume the changes of weights across $\mathbf{u}$, not the change of causal graph structure. As we mentioned in section 3.2, the main challenge of non-identifiablity is due to transitivity. From this point, existing methods, including sparse graph structure and temporal information, could be regarded as feasible ways to handle the transitivity. Therefore, allowing the change of weights is not necessary, but a sufficient condition to handle the transitivity. Please see our new report about the changes of part of weights in section A.6 for further discussions. Currently, there are three primary approaches to achieve identifiability as mentioned in Introduction, and all these approaches have limitations, to a certain extent, as we summarised in section Related Work. This motivates us to propose a new research line in the hope that it might assist in taking a step forward in causal representation learning.

---

> > ### Comment · Reviewer_kexo · 2022-12-02
> > **Response**
> >
> > Thank you for your clarification and revision of the paper. I have increased my scores accordingly.
> >
> > - I agree that Corollary 4.2 is about theoretical identifiability, and it does not rely on the estimation algorithm and the chosen threshold. The question in my original review was more focused on how Corollary 4.2 justifies your estimation method. Regarding the comment "score-based methods also find the structure based on independence relations implicitly", I am not sure if this is the right way to put it; since in your case reading-off independence relations/graph structures is more or less dependent on your subjective perception of the weight scaling (i.e., threshold), which will certainly lead to robustness issues. General score-based methods of course have similar estimation issues but working with latent confoundings have made this particularly outstanding. I would recommend at least include more empirical sensitivity analysis on this issue.
> > - And just to clarify my concerns regarding "varying graphs", when the weights are changing, wouldn't it also imply that given a fixed threshold, varying weights will give varying graphs?

---

> > > ### Author Response · Authors · 2022-12-03
> > > **Thank you again for your invaluable feedback and A kind request for double check for the update of score**
> > >
> > > Dear kexo,
> > >
> > > Your valuable comments have helped improve our presentation a lot.
> > >
> > > We will include more empirical sensitivity analysis in the final version. And, we agree that the graphs may be varying, since the weights totally depend on u, which implies the change of graphs across u.
> > >
> > > A kind request for double check for the update of score: If we remember correctly, the original score is 6 (Please let us know, if we misremember), but your comments mentioned "I have increased my scores accordingly." .
> > >
> > > Thank you once again for your time devoted to this paper.

---

### Official Review · Reviewer_4oGs · 2022-10-26

**Confidence:** 5
**Correctness:** 2
**Technical Novelty And Significance:** 2
**Empirical Novelty And Significance:** 2
**Recommendation:** 3

**Clarity, Quality, Novelty And Reproducibility:**

The paper is clearly written. However, some important things related to the method for learning the MEC are missing as I highlight in the weaknesses above. The work needs significant revision but it is promising. I do not see the code to reproduce the experiments.

**Strength And Weaknesses:**

**Strengths**
The authors study a very important problem and propose an interesting approach that extends works such as causal VAE. A main limitation of causal VAE is that the each latent is independent of the rest conditional on one component of the auxiliary information vector. To address this limitation, authors propose a new data generation model, the weight variant linear causal model. The weight variations in the linear causal model allow the authors to achieve identification. These weight variations are connected to interventions, which makes the setup interesting.

**Weaknesses**

I have several concerns that I highlight below.

1. **Regarding the weight variant causal model** The new data generation process considered by the authors seems quite artificial to be honest. Having access to an auxiliary variable that impacts all weights in a linear SEM seems too much of a stretch. Do authors have some good examples that show otherwise? If yes, then authors should provide a convincing case of such examples during the rebuttal. My understanding tells me that data generation process seems to have been reverse engineered to fit the requirements of the proof. In other words, DGP has been engineered to allow for structural dependency but at the same time leverage as much of iVAE as possible.

2. **Regarding Corollary 4.2** The paper leverages the work of Huang et al. to achieve identification of causal DAG up to the Markov equivalence class. Since this is quite important to the paper the authors should have fully explained if the assumptions that are required in Huang et al. are fully met. Also, the authors should have described the exact steps of the procedure to go from Markov equivalence of z and u to z only. If this was excluded due to space constraints, I am a little surprised as this seems quite important. I also do not see a discussion in depth on this in the Appendix.
Regarding the proof of Corollary 4.2. The proof of the corollary is incomplete. The authors talk about recovery of MEC of z union u. That can be done from good old works of Pearl and Verma that date back to 1990. How to go from there to the MEC of z?


3. **Regarding experiments** In the set of experiments that are carried out on FMRI data. I do not see a description of what is the auxiliary information. In the absence of auxiliary information how does the framework of authors even kick in. Further, the authors seem to learn the DAG just by thresholding the weights learned. I was expecting to see a set of DAGs that are Markov equivalent and some of them would correspond to the true one.
Also, to validate Corollary 4.2 the authors should have carried out synthetic data based experiments to show that indeed the MEC is recovered.

4. **Role of weight variation model** Since weight variation is the key part of the paper. Some ablation experiments that explain how different choices for weight variation model help in different types of identification would have been useful. In the current form the model carries out global interventions and that is not realistic. What would happen if the interventions were localized, i.e., some components of u only interact with some edges, do we get partial identification at least?




**Summary Of The Paper:**

In causal discovery, one starts with input variables and then learns a causal graph describing the relationships between the input variables. In recent years, a complementary question, which asks "how to learn causal variables from low-level raw data (e.g., images)?" has gained attention and this task is often termed "causal representation learning". Many recent works in the area either leverage simplistic assumptions on the underlying causal DAG (e.g., independence of latents conditional on observed auxiliary variable in iVAE). Other works leverage temporal data to deal with more general causal DAGs (e.g., CITRIS). In this work, the authors tackle the question without making assumptions on the structure of the underlying causal DAG. To tackle the question, the authors assume a linear structural equation model that governs interactions between the latents and also require to observe an auxiliary variable that impacts the weights of the weights of the structural equation model. Under the above assumptions, the authors extend the theory from iVAE to achieve identification in the proposed setting. Having shown identification of the latents the authors show that the causal DAG of the latents can be identified up to the Markov equivalence class. Lastly, authors propose an extension of the iVAE based procedure by adapting the prior based on the linear SEM that is assumed. The authors show that the procedure is effective in synthetic and semi-synthetic experiments.

**Summary Of The Review:**

The work proposes a new DGP to tackle the question of causal representation learning. The DGP proposed seems to be quite contrived and that is one major limitation. The proof of the theorems are correct, but the proof of the corollary is incomplete or has details missing.

---

> ### Author Response · Authors · 2022-11-16
> **Rebuttal**
>
> __*Q1: Regarding the weight variant causal model.*__
> Latent causal discovery is a notoriously hard problem that requires fundamental theoretical advances before practical algorithms can be developed. Identifying latent causal graph is generally not possible without certain assumptions. This work proposes a new research line to explore the relation between change of causal influences and identifiability of latent causal discovery. Experiments on simulation data and fMRI data verify the theoretical results. We believe that this might be a significant source of inspiration, and promote causal representation learning in real applications.
>
> __*Q2: Corollary 4.2, a) About assumptions.*__
> Corollary 4.2 relies on the Markov condition and faithfulness assumption and the assumption that the latent change factor (i.e., causal strength in the linear case) can be represented as a function of the domain index $u$. Hence, it relies on the same assumptions as that in Huang et al. We have emphasized this point in the appendix.
>
> __*Q2: Corollary 4.2, b) Exact steps from MEC of z and u to z only.*__
> Denote by $M$ the Markov equivalence class over $z \cup u$, and by $M_z$ the Markov equivalence class over $z$. Then after removing variable $u$ in $M$ and its edges, the resulting graph (denoted by $M’_z$) is the same as $M_z$. This is because of the following reasons. First, it is obvious that $M’_z$ and $M_z$ have the same skeleton. Second, in this paper, $u$ has an edge to every $z_i$ when considering $n$ as latent noise variables, because all causal strength and noise distributions change with $u$. Hence, there is no v-structure over $u$, $z_i$, and $z_j$, so it is not possible to have more oriented edges in $M'_z$. Therefore, $M’_z$ and $M_z$ have the same skeleton and the same directions.
>
> __*Q2: Corollary 4.2, c) Old works of Pearl and Verma.*__
> The reason that we use the results from Huang et al. is that here actually $u$ is treated as a surrogate variable to capture latent changing factors $\lambda(u)$ (note that the actual variable $\lambda(u)$ is unobserved) that leads to the heterogeneous data, similar to the setting in Huang et al, while the old works assume all variables are observed and the data is i.i.d.
>
> __*Q3: Experiments: a) the auxiliary information of fMRI.*__
> As we mentioned, fMRI data contains signals on the same person in 84 successive days, here different days can be regarded as the auxiliary information. We have clarified this point in the new rebuttal version.
>
> __*Q3: Experiments: b) Learn the DAG just by thresholding the weights learned.*__
> Despite our identifiability results, some challenges may often appear in reality, e.g., finite sample size and local optimization. Given these, we may need to refine our estimated graphs by thresholding the learned weights to obtain final graphs. Sparsity based regularization (e.g., $L_1$) on weights may effectively replace the threshold process. In this work we mainly focus on identifiability results, designing more effective methods than the proposed method in this work to discover weight-variant latent causal graph is interesting in the future work.
>
> __*Q3: Experiments: c) Experiments to show that indeed the MEC is recovered.*__
> To validate Corollary 4.2, we report SHD of the recovered completed partial directed acyclic graphs by the proposed method, as shown in Figure 5 (c) in experiments. We sincerely thank the reviewer for this suggestion.
>
> __*Q4: Role of weight variation model: Ablation experiments, local changes.*__
> Interesting question. As a first work to discuss the change of causal influences for identifiablity, we pay close attention to changes of global weights. Considering the changes of part of weights may relax our assumptions and provide new insights in theory. We report new experimental results to give further discussions for the local changes, please see appendix A.6 for details.

---

> > ### Author Response · Authors · 2022-12-04
> > **Could you kindly check whether our response properly addressed your concerns?**
> >
> > Dear Reviewer 4oGs
> >
> > We appreciate your time devoted to reviewing the manuscript. We have provided responses to your comments and an updated submission.
> >
> > Could you please check whether they properly addressed your concern? Your feedback would be appreciated.
> >
> > Thank you very much!

---

> > > ### Comment · Reviewer_4oGs · 2022-12-05
> > > **Thanks for feedback**
> > >
> > > Hello
> > >
> > > I have read your rebuttal. Thanks for the clarifications, especially regarding the experiment. I am still not convinced regarding your argument to support the modeling assumptions. I also see there are some serious concerns raised by Reviewer Faed and that has even led to major changes in the paper. Unfortunately, the paper is not ready for acceptance and I am decreasing my score.

---

### Author Response · Authors · 2022-11-16
**General Response**

We thank the reviewers for their valuable feedbacks and time devoted to our work.

Reviewer iGWL concisely conveyed our contribution: ``1, an additional latent causal variables layer is generated conditioned on the latent noise variables; 2, the latent causal variables are modelled via linear Gaussian models with -dependent causal weights. Under certain conditions, the authors shows that 1), the latent causal variables can be identified up to permutation and linear scaling; 2), the causal graph among latent causal variables can be identified up to its Markov equivalence class. The authors then proposed practical algorithms for identifying the latent causal variables and conducted both synthetic and real-world experiments."

Most of reviewers are positive in writing, novelty, importance of the manuscript. Writing: the Reviewers, 4oGs, kexo, 2uCg and P5md, consider that 'The paper is clearly written (4oGs)', 'This paper is very  well-written (kexo)', 'This paper is very well and claerly written (2uCg)'. Novelty: 'the authors study a very important problem and propose an interesting approach (4oGs)', 'This is an interesting paper that addresses an important question in causal representation learning (kexo)', 'the quality is good. Also, it contains novel theories on the problem (2uCg)', 'Well written paper with potentially some interesting results on latent model identifiability (P5md)',  'identifiability result introduced in this work appears to be novel and very interesting (FaeD)'.

Some reviewers, including '4oGs', 'kexo', 'DXAx' and 'P5md', raised a concern about the model assumptions, we have addressed it in the individual response, and by adding an additional section (see A.2) for further understanding assumptions in Theorem 4.1.

Some reviewers, including '4oGs', 'P5md' and 'FaeD', raised a concern on the additional experiments and experimental details, which are addressed by adding an additional experiments and implementation details (see more details in section A.5, A.6 and A.7).

Reviewer `FaeD' raised a concern about the details of Eq. (42) in the original version, which is addressed by adding a new proof step (step IV in the new rebuttal version).

We have supplied additional experimental results following the reviewers' suggestions and/or requests, and answered all questions (see more details in the individual responses).

---

### Decision · Program_Chairs · 2023-01-20

**Decision:**

Reject

**Justification For Why Not Higher Score:**

The proof of the core technical result has some steps that are not fully transparent. Thus the paper is not ready for acceptance as is.

**Justification For Why Not Lower Score:**

N/A

**Metareview: Summary, Strengths And Weaknesses:**

This work is about causal representation learning without assuming causal DAG structure. It assumes linear models for the latents and achieves identification of the DAG (up to Markov equivalence classes) via additional auxiliary variables.

strength

+ The problem is important and the weight variant linear causal model is interesting for achieving identification.

weakness

+ The requirement of an auxiliary variable impacting all weights of a linear SEM could be potentially stringent.

+ Some crucial steps in the proof of Theorem 4.1 are not completely clear. During the rebuttal period, the paper underwent substantial changes in this proof and introduced new assumptions. However, the new assumption appears to be quite strong, and the use of Sorrenson et al. (2020)'s result in the proof remains unclear and not fully transparent.

We encourage the authors to make the proof more transparent for the identification result and submit to a future venue.


**Summary Of Ac-Reviewer Meeting:**

N/A